# Ice nucleating particles from multiple aerosol sources in the urban environment of Beijing under mixed-phase cloud conditions

Cuiqi Zhang[1], Zhijun Wu[1,2], Jingchuan Chen[1], Jie Chen[1,a], Lizi Tang[1], Wenfei Zhu[1], Xiangyu Pei[4], Shiyi Chen[1], Ping Tian[5], Song Guo[1], Limin Zeng[1], Min Hu[1], Zamin A. Kanji[3]

[1]State Key Joint Laboratory of Environmental Simulation and Pollution Control, College of Environmental Sciences and Engineering, Peking University, Beijing, 100871, China
[2]Collaborative Innovation Center of Atmospheric Environment and Equipment Technology, Nanjing University of Information Science and Technology, Nanjing, 210044, China
[3]Institute for Atmospheric and Climate Science, ETHZ, Zurich, 8092, Switzerland
[4]College of Environmental and Resource Sciences, Zhejiang University, Hangzhou, 310058, China
[5]Beijing Weather Modification Center, Beijing, 100089, China
[a]now at: Institute for Atmospheric and Climate Science, ETHZ, Zurich, 8092, Switzerland

*Correspondence to*: Zhijun Wu (zhijunwu@pku.edu.cn)

**Abstract.** Ice crystals occurring in mixed-phase clouds play a vital role in global precipitation and energy balance because of the unstable equilibrium between co-existent liquid droplets and ice crystals, which affects cloud lifetime and radiative properties, as well as precipitation formation. Satellite observations proved that immersion freezing, i.e., ice formation on particles immersed within aqueous droplets, is the dominant ice nucleation (IN) pathway in mixed-phase clouds. However, the impact of anthropogenic emission on atmospheric IN in the urban environment remains ambiguous. In this study, we present in situ observations of ambient ice nucleating particle number concentration ($N_{INP}$) measured at mixed-phase cloud conditions (-30 ℃, relative humidity with respect to liquid water $RH_w$ = 104%) and the physicochemical properties of ambient aerosol, including chemical composition and size distribution, at an urban site in Beijing during the traditional Chinese Spring Festival. The impact of multiple aerosol sources such as firework emissions, local traffic emissions, mineral dust and urban secondary aerosols on $N_{INP}$ is investigated. The results show that $N_{INP}$ during the dust event reaches up to 160 # $L^{-1}$, with an activation fraction (AF) of 0.0036% ±0.0011%. During the rest of the observation, $N_{INP}$ is on the order of $10^{-1}$ to 10 # $L^{-1}$, with an average AF between 0.0001 to 0.0002%. No obvious dependence of $N_{INP}$ on the number concentration of particles larger than 500 nm ($N_{500}$) or black carbon (BC) mass concentration ($m_{BC}$) is found throughout the field observation. The results indicate a substantial $N_{INP}$ increase during the dust event, although the observation took place at an urban site with high background aerosol concentration. Meanwhile, the presence of atmospheric BC from firework and traffic emissions, along with urban aerosols formed via secondary transformation during heavily polluted periods do not influence the observed INP concentration. Our study corroborates previous laboratory and field findings that anthropogenic BC emission has a negligible effect on $N_{INP}$, and that $N_{INP}$ is unaffected by heavy pollution in the urban environment under mixed-phase cloud conditions.

## 1 Introduction

Ice crystals in clouds can form via homogeneous freezing of aqueous droplets below -38 ℃, or via heterogeneous ice nucleation (IN) with the aid of foreign interfaces offered by atmospheric ice nucleating particles (INPs) through immersion/contact freezing of existing droplets at higher temperature or direct deposition/condensation of water vapor below water saturation (Pruppacher and Klett, 1997; Vali et al., 2015; Kanji et al., 2017). Mixed-phase clouds occur where super-cooled liquid water droplets co-exist with ice crystals and are normally sustained between -38 and 0 ℃ in the atmosphere, with ice melting rapidly at warmer temperature and droplets freezing homogeneously at colder temperature (Boucher et al., 2013; Korolev et al., 2017). The Wegener–Bergeron–Findeisen process in mixed-phase clouds favors ice crystal growth at the cost of liquid droplet evaporation (Wegener, 1911; Bergeron, 1935; Findeisen, 1938), leading to ice water content and ice crystal size change, which further result in changes of mixed-phase cloud lifetime and radiative properties, as well as global precipitation pattern (Cantrell and Heymsfield, 2005; Field and Heymsfield, 2015; Mülmenstädt et al., 2015; Korolev et al., 2017; Heymsfield et al., 2020). Satellite observations demonstrate that the predominant ice formation pathway in mixed-phase clouds is immersion freezing (e.g., Ansmann et al., 2008; de Boer et al., 2011; Silber et al., 2021). In this mode, INPs immersed within super-cooled aqueous droplets provide an interface that decreases the liquid-solid phase transition energy barrier and aids droplet freezing (Pruppacher and Klett, 1997; Vali et al., 2015; Kanji et al., 2017).

Most of the particles in highly populated urban areas originate from local emissions, including ground transportation, cooking, coal and biomass burning, leading to significant production of carbonaceous particles, including organic compounds and elemental carbon, as well as inorganic salts (e.g., Sun et al., 2016). Apart from local emissions, regional transportation, such as transportation of mineral dusts and pollutants from adjacent areas, also contributes significantly to urban particle population under appropriate meteorological conditions, during which aging can significantly modify particle physicochemical properties, such as chemical composition, morphology, and mixing state (e.g., Lin et al., 2016; Sun et al., 2016; Hua et al., 2018; Zhang et al., 2020b; Lei et al., 2021; Li et al., 2021). Previous studies have confirmed that several kinds of atmospheric particles, including mineral dusts, carbonaceous particles, and biological species, can act as immersion INP (Hoose and Möhler, 2012; Murray et al., 2012; Kanji et al., 2017 and references therein). When present at atmospherically relevant amounts in droplets, mineral dusts mostly catalyze super-cooled aqueous droplet freezing below -15 ℃ (Hoose and Möhler, 2012 and references therein; Murray et al., 2012; Kanji et al., 2017 and references therein), while biological species, such as pollen, fungal spores, and viruses, generally exhibit immersion IN activity below -5 ℃ and are fully activated below -10 ℃ to -20 ℃ (e.g., Chou, 2011; Conen et al., 2015; Polen et al., 2016; Kanji et al., 2017; Conen et al., 2022; Porter et al., 2022). The reported atmospheric immersion INP number concentration ($N_{INP}$) were measured between -5 ℃ and -38 ℃ and were normally on the orders of $10^{-2}$ to $10^3$ # L$^{-1}$ (e.g., Rogers et al., 1998; DeMott et al., 2010; Chen et al., 2018; Porter et al., 2022).

Among all types of airborne particles, mineral dusts are commonly acknowledged as a major source of effective atmospheric immersion INPs (e.g., DeMott et al., 2003; Archuleta et al., 2005; Kanji and Abbatt, 2006; Welti et al., 2009; DeMott et al., 2010; Atkinson et al., 2013; Cziczo et al., 2013; DeMott et al., 2015; Chen et al., 2021). It was also reported

that crystalline ammonium sulfate could nucleate ice heterogeneously below water saturation (Abbatt et al., 2006). But the effectiveness of carbonaceous particles and inorganic salts acting as INP under mixed-phase cloud conditions remains elusive (Schill et al., 2016; Chen et al., 2018; Kanji et al., 2020; Schill et al., 2020; Wolf et al., 2020). Although certain types of black carbon (BC) and organic particles exhibited INP activity at temperatures below -38 ℃ through deposition IN (Murray et al.,

2010; Mahrt et al., 2018; Nichman et al., 2019; Zhang et al., 2020a), field observations (Chen et al., 2018; Adams et al., 2020) and laboratory experiments (Schill et al., 2016; Kanji et al., 2020; Schill et al., 2020) suggest that carbonaceous particles might not affect ice crystal formation via immersion mode. Besides, organic coatings are likely to impede carbonaceous particles from acting as effective deposition INP at temperatures below -38 ℃ (Nichman et al., 2019; Zhang et al., 2020a).

Previous modelling work confirmed that anthropogenic INP emission could alter the size of ice crystals in clouds and

change cloud lifetime and global precipitation pattern (Zhao et al., 2019). Yet, there is limited published direct evidence on the contribution of anthropogenic particles to ice crystal formation in highly populated areas (Knopf et al., 2010; Corbin et al., 2012; Chen et al., 2018; Che et al., 2019; Che et al., 2021). Knopf et al. (2010) used filter samples collected from a highly populated urban area in Mexico City and an optical IN microscopy technique to report that anthropogenic particles dominated by organic components might catalyze ice formation well below water saturation at temperature below -38 ℃. Such organic-

rich anthropogenic particles also demonstrated ice formation potential via immersion pathway above -38 ℃ in their study. Corbin et al. (2012) suggested that coupling atmospheric dust, elemental carbon, and biomass burning particle concentration together could provide the best estimation for atmospheric INP concentration in downtown Toronto at -34 ℃ just below water saturation, but the share of each particle category remained unclear due to limited data. Chen et al. (2018) quantified off-line immersion INP concentration between -6 ℃ and -25 ℃ using filter samples collected every 12 hours during a heavily polluted

2016 wintertime in Beijing. Even though a high level $PM_{2.5}$ with complex chemical composition was sampled during the pollution period in the urban area, these aerosols did not act as superior INPs, and the highest INP concentration measured at -25 ℃ was below 10 # $L^{-1}$, similar to what was observed in remote regions such as the Swiss Alps (Boose et al., 2016a; Lacher et al., 2017). The INP concentration reported by Chen et al. (2018) was insensitive to particle number concentration and particle chemistry in an atmosphere that was dominated by anthropogenic emissions. The absence of a correlation between immersion

INP concentration and particle number during a pollution period was further supported by Bi et al. (2019) in an online immersion INP concentration field observation at an suburban site in Beijing during May to June, 2018, using a continuous flow diffusion chamber (CFDC) operated above water saturation between -20 ℃ to -30 ℃. The lack of online particle chemistry information impedes aerosol source correlation in these studies (Chen et al., 2018; Bi et al., 2019). Che et al. (2019) reported a positive correlation between the total atmospheric INP concentration and air pollution degree during springtime in

Beijing. INP concentration was measured by a Bigg-mixing cloud chamber for one month in 2017, and the total atmospheric INP concentration could reach 1500 # $L^{-1}$ at -30 ℃ (Che et al., 2019; 2021).

Currently, a knowledge gap still exists on the magnitude and dominant source of ambient INPs in highly populated urban area, as well as the dependence of INP concentration on anthropogenic particle emission, hampering the estimation of global

atmospheric INP concentrations (Boucher et al., 2013; Seinfeld and Pandis, 2016). In this paper, we report the in situ INP
concentration measured at mixed-phase clouds condition (-30 ℃, relative humidity with respect to liquid water of 104% $RH_w$
= 104%) during the traditional Chinese Spring Festival at an urban site in Beijing. Urban particle emission sources were
distinguished based on the online chemical characterization using an Aerosol Chemical Speciation Monitor (ACSM). The
correlations between immersion INP concentration, meteorology condition, and aerosol physiochemical properties are also
explored.

## 2. Methods

### 2.1 Sampling

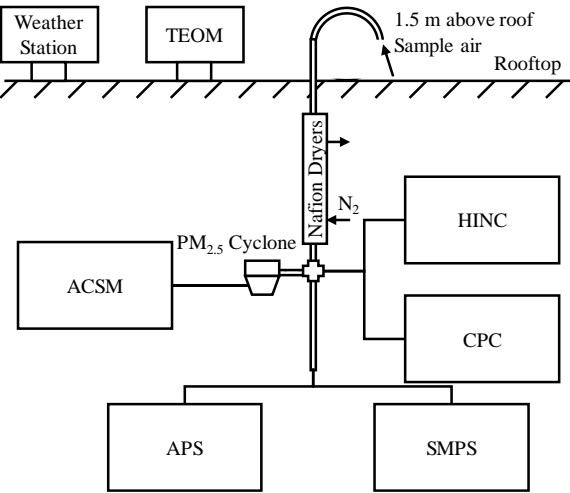

**Figure 1: Schematic of the sampling and experimental setup (not drawn to scale).**

The sampling site (39 °59′20″N, 116 °18′26″E) is located on the roof of a six-floor building (~30 m above ground level) at
Peking University, which is adjacent to the north-western 4th ring road of Beijing. The site lies about 250 m west of a busy
street with heavy traffic. At the sampling site, meteorological parameters, including wind speed, wind direction, $RH_w$, and
temperature, were measured by a weather station (MetOne Inc.). The mass concentration of particulate matter (PM) with
aerodynamic diameter ($d_a$) smaller than 2.5 and 10 μm ($PM_{2.5}$ and $PM_{10}$, respectively) were measured by a tapered element
oscillating microbalance (TEOM) monitor. The temporal resolutions of meteorology and PM data were 1 minute.

Ambient air was sampled through a stainless tube with an inner diameter of 12.7 mm. The tube inlet was arched so that
it was bent facing downwards (see Fig. 1) to prevent water contamination, with $d_{10}$ and $d_{50}$ (the $d_a$ at which 10% and 50%
particle could be transported to instruments through the inlet, respectively) being ~20 μm and 13.4 μm, respectively
(Brockmann, 2011, eq. 6-23 to 6-29). The sample flow was then split and pumped into different instruments. The relative

humidity of the sample stream ($RH_{w, sample}$) was kept below 2% by passing through two consecutive 47 cm Nafion™ dryers (Perma Pure, LLC.) using 4 LPM nitrogen as sheath gas during the experiment. A schematic of the setup is shown in Fig. 1.

## 2.2 Instrumentation

### 2.2.1 Particle number size distribution

Sub-micron particle number size distribution was measured by a scanning mobility particle sizer (SMPS, model 3082, comprising a 3082 classifier, a 3081 long DMA, and a 3776 CPC; TSI Inc.). The sampling flow rate of SMPS was set to 0.3 LPM with a sheath-to-sample ratio of 10:1, resulting in an electrical mobility size range from 14.6 nm to 710.5 nm. An impactor was installed at the DMA inlet to remove particles larger than 735 nm. Furthermore, multiple charging correction was applied when processing the sub-micron particle number size distribution.

An aerodynamic particle sizer (APS, model 3021; TSI Inc.) was used to provide number size distribution for ambient aerosols with $d_a$ ranging from 0.542 μm to 19.81 μm every minute. The aerodynamic particle number size distribution obtained from APS could be converted to particle mobility size ($d_m$) distribution by assuming the effective density of ambient particles to be 1.5 g cm$^{-3}$, which is commonly used for urban atmosphere (Khlystov et al., 2004; Chen et al., 2018; Qiao et al., 2018; An et al., 2019). The sampling flow rate of APS was 1 LPM.

### 2.2.2 Particle chemical composition

Real-time non-refractory PM$_1$ ($d_a$ smaller than 1.0 μm) mass loading and chemical composition was measured by an ACSM (Aerodyne Inc.) equipped with a quadrupole analyzer (Ng et al., 2011). The sampling flow rate of ACSM was 0.1 LPM. A PM$_{2.5}$ cyclone was installed upstream ACSM inlet to prevent inlet clog by particles with $d_a$ larger than 2.5 μm. The 50% transmission efficiency range of ACSM is ~60-660 nm (Liu et al., 2007). The time resolution of an ACSM scan was set to 15 minutes. Meanwhile, BC mass concentration was monitored by a multi-angle absorption photometer (MAAP, model 5012; Thermo, Inc.) with a temporal resolution of 1 minute.

### 2.2.3 Ice nucleating particle (INP) concentration

In situ immersion INP concentration was measured by a Horizontal Ice Nucleation Chamber (HINC) at fixed lamina condition throughout the observation period, i.e. with a lamina temperature ($T_{lam}$) of -30 ±0.2 ℃ and $RH_w$ = 104 ±2.2% (equivalent to $RH_i$ = 140 ±3.0%, where the subscript i denotes ice). HINC is a CFDC type instrument made of two flat parallel copper plates. The temperature of each plate is controlled independently to create supersaturation along the chamber centerline lamina. To minimize the impact of convection, the top plate of HINC is warmer than the bottom plate. Ice crystal size and number was measured by a 6-channel optical particle counter (OPC; MetOne Inc.). The injector position, and thus the flow structure of HINC in this study is identical to the setting of Lacher et al. (2017). Estimated particle gravitational settling within HINC is consistent with the OPC measurement by Lacher et al. (2017), i.e. ~23.5% for 1 μm particle, 46.6% for 1.5 μm particles, and 100% for particles larger than 5 μm (Brockmann, 2011, eq. 6-51 to 6-53). Therefore, only particles larger than 5 μm detected

by the HINC OPC are counted as ice crystals. The INP concentration measurements in this study are representative of ambient particles smaller than 1.5 μm. For more detailed HINC design and operating principle information, please refer to Lacher et al. (2017) and Kanji and Abbatt (2009). In addition, the gravitational settling estimated for 1 μm, 2 μm, and 5 μm particles in the tubing upstream HINC inlet are estimated to be ~2.1%, 7.7%, and 41.0%, respectively (Brockmann, 2011, eq. 6-51 to 6-53). Such sampling line loss is considered to be negligible because particles larger than 1 μm were scarce during the observation
(Table 1).

     In this study, both warm and cold walls of HINC lined with glass-fiber filter paper were wetted with ~150 mL de-ionized water each day before the experiment start, or after running experiment for 4 hours. After draining for ~15 min, the wall temperatures of warm and cold walls would be set to -21.2 and -38.8 ℃ respectively to achieve desired lamina temperature (-30 ℃) and $RH_w$ (104%). The sampling flow rate of HINC was 0.26 LPM, surrounded by 2.57 LPM particle-free nitrogen
sheath gas. During the experiment, sampling air would pass through a particle filter for 5 minutes after every 15 minutes of measurement to quantify HINC background count detected by the OPC. HINC background counts follow a Poisson distribution, based on which the average background count is determined. Average ice crystal concentration (equivalent to $N_{INP}$) of the 15-minute measurement period is calculated by firstly subtracting the average background particle counts from measurement counts, and secondly converting particle counts to number concentration using HINC sampling flowrate. The subtraction might
produce negative $N_{INP}$ when the signal of OPC during the measurement is undistinguishable from background noise. Therefore, this study reports positive $N_{INP}$ as is, and replaces negative $N_{INP}$ with the minimum quantifiable concentration of OPC (0.26 # $L^{-1}$) following the method in Lacher et al. (2017). Ambient particle number concentration entering HINC was monitored by a CPC (Model 3775, $d_{50} = 4$ nm; TSI Inc.) connected in parallel with HINC at the aerosol inlet (see Fig. 1).

     Activation fraction (AF) and ice-active surface site density ($n_S$) were selected as IN activity parameters in this study (Vali
et al., 2015; Kanji et al., 2017). AF is the ratio between ice crystal number concentration at HINC outlet (calculated from OPC counts, as stated above) and total particle number concentration at HINC inlet (measured by CPC). $n_S$ is defined as the number of ice-active surface sites per unit surface area of INPs, and allows IN activity inter-comparison between different aerosol species and different studies as a normalized parameter (Hoose and Möhler, 2012; Vali et al., 2015). In this study, total surface area $S$ for poly-disperse ambient particles was firstly derived by assuming particles to be spherical and integrating the particle
mobility size distribution (Lacher et al., 2018; Bi et al., 2019; Chen et al., 2021). Dividing $N_{INP}$ by $S$ yields $n_S$ (Connolly et al., 2009; Hoose and Möhler, 2012; Niemand et al., 2012; Vali et al., 2015; Lacher et al., 2018; Bi et al., 2019).

## 3. Results and discussion

### 3.1 Overview

The observation lasted from Feb. 10[th] to 28[th], overlapping with the traditional Chinese Spring Festival for the year 2021. Figure
2 displays (a) the chemical composition of non-refractory $PM_1$ mass concentration and (b) particle number size distribution.

Figure 3 shows the ambient meteorological conditions, including (a) wind speed and wind direction, (b) temperature and $RH_w$, (c) PM$_{2.5}$ and PM$_{10}$ mass concentrations and $N_{INP}$. Fig 4 presents the BC mass concentration ($m_{BC}$) variation during the observation in addition to $N_{INP}$. The grey shading in Fig. 2-4 indicates the IN experiment time periods.

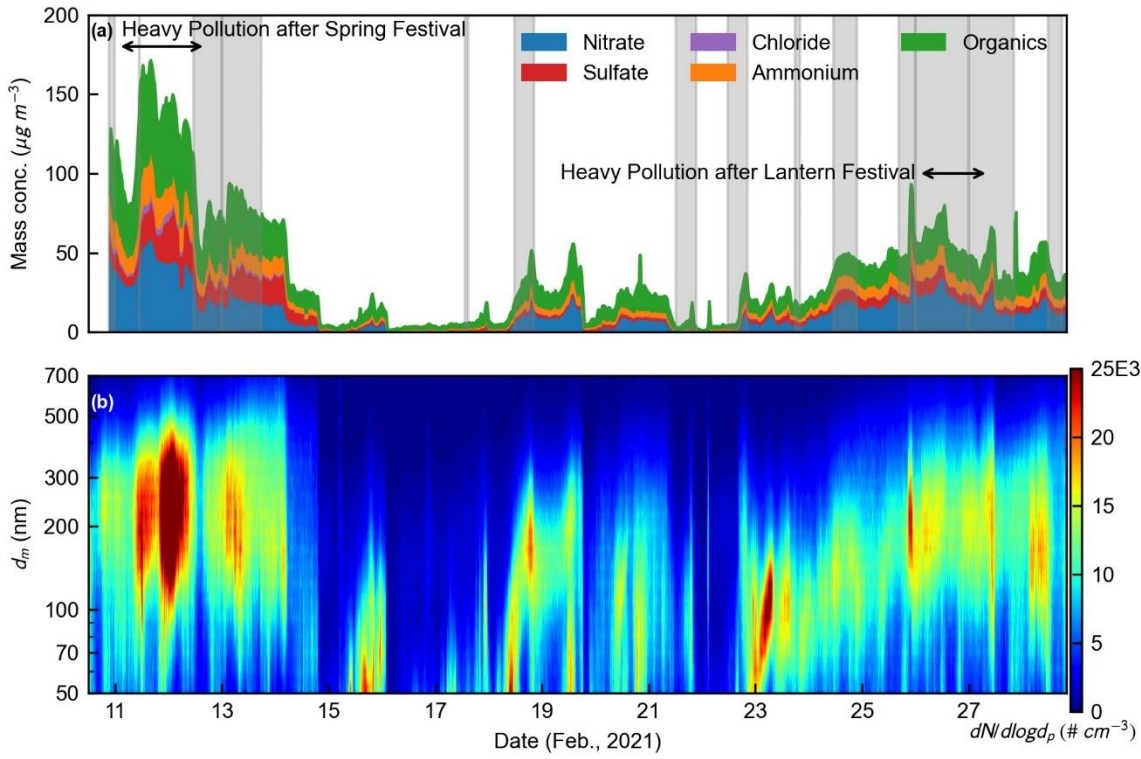

Figure 2: Time series of (a) non-refractory PM$_1$ mass concentration and (b) particle number size distribution. The grey shading and black arrows in the upper panel indicate IN experiment time periods and heavy pollution after celebrations, respectively.

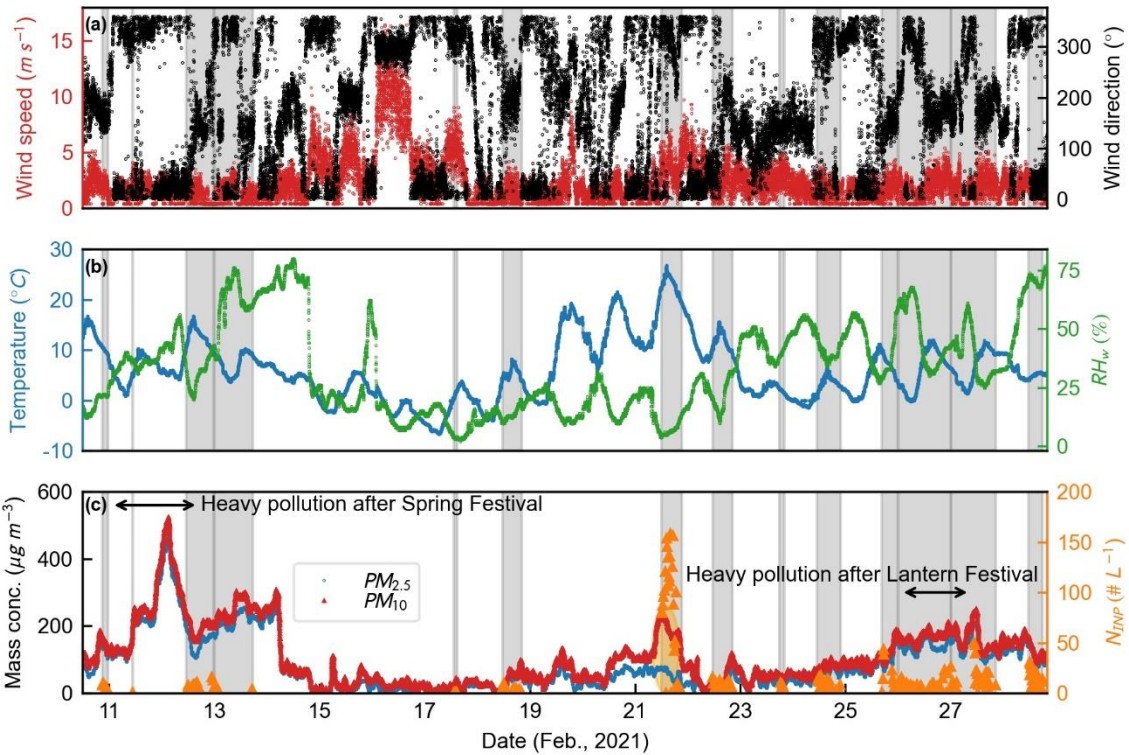

**Figure 3: Times series of (a) wind speed and direction, (b) ambient temperature and relative humidity with respect to liquid water ($RH_w$), (c) PM$_{2.5}$ and PM$_{10}$ mass concentrations and immersion INP number concentration ($N_{INP}$). The grey shading in each panel**
**indicates IN experiment time periods; the orange shading and arrows in the lower panel mark the dust event on Feb. 21$^{st}$, 2021 and the heavy pollution after celebrations, respectively.**

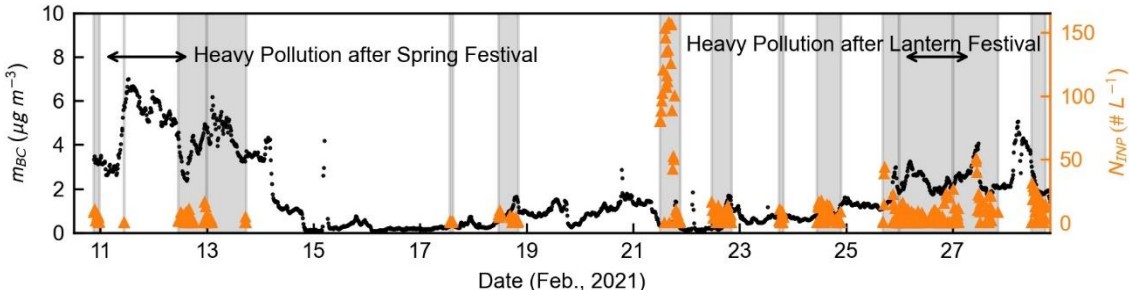

**Figure 4: Time series of black carbon (BC) particle mass concentration ($m_{BC}$) and $N_{INP}$. The grey shading and black arrows indicate IN experiment time periods and heavy pollution after celebrations, respectively.**

On Feb. 11$^{th}$ and 26$^{th}$ night, fireworks were lit for celebrations of the Spring and Lantern Festivals. Large amounts of particles emerged after the celebrations due to firework emission, as indicated by the sharp growth of non-refractory PM$_1$ mass concentration comprising organic components and chloride salts in Fig. 2a. Up to 7 µg m$^{-3}$ of BC particles mainly originating from firework emission were also detected during and after the celebrations, as illustrated in Fig. 4. Apart from the firework emission, the mass concentration of nitrate and sulfate also increased significantly (Fig. 2a) under the relatively stagnant and

humid meteorological conditions after Feb. 11th and 26th nights (Fig. 3b), indicating secondary pollutant formation (e.g., Wu et al., 2018). Such secondary pollutant transformation processes produce particles in the accumulation mode, as illustrated by the high level of particle concentration peaking between 200-300 nm in Fig. 2b. During heavy pollution after the Spring and Lantern Festivals, $PM_{2.5}$ mass concentration respectively approached ~550 and 200 μg m$^{-3}$ (Fig. 3c). The synergetic heavy pollution induced by secondary pollutant formation and firework emission are marked with arrows in Fig. 2-4.

On Feb. 18th, the first workday after the Spring Festival holiday (Feb. 11th to Feb. 17th), people swarmed into downtown Beijing and the mass concentrations of non-refractory $PM_1$ and $PM_{2.5}$ increased during rush hours, as can be seen in Fig. 2a and 3c, respectively. $m_{BC}$ also increased because of increasing use of passenger vehicles during rush hours on workdays after Feb. 18th, as shown in Fig. 4.

During the whole observation, there was minor difference between ambient $PM_{2.5}$ and $PM_{10}$ mass concentrations ($PM_{10-2.5}$) except for Feb. 21st afternoon (Table 1), when there was a significant increase of $PM_{10}$ mass concentration as highlighted by the orange shading in Fig. 3c, implying that large particles with $d_a$ ranging between 2.5 μm and 10 μm were present, which were most likely to be mineral dust particles (e.g., Park et al., 2004; Hoffmann et al., 2008; Rai et al., 2021). Besides, aerodynamic particle number size distribution exhibited a clear shift towards the larger end during the dust event (Fig. A1) with a mode size around 1 μm, which further confirmed the presence of large particle during the dust event. Aerosol optical depth (AOD) derived from MODIS Aqua Deep Blue Collection 6 dataset at 550 nm (Acker and Leptoukh, 2007) also shows elevated aerosol loading on Feb. 21st afternoon at the sampling site compared to Feb. 20th (Fig. A2). Based on measured particle mass concentration level and meteorological conditions, the observation days are categorized into different scenarios, i.e. dust event, clean, and heavy pollution days, as summarized in Table 1.

**Table 1 The date, number concentrations of immersion INP ($N_{INP}$) and ambient particle ($N_{CPC}$), activation fraction (AF), ice-active surface site density ($n_S$), number concentrations of particles ranging from 500 nm to 1.5 μm ($N_{500}$) and 1000 nm to 1.5 μm ($N_{1000}$), as well as mass concentrations of BC particles ($m_{BC}$) and ammonium salt ($m_{ammo}$), $PM_{2.5}$ and $PM_{10}$, and $PM_{10-2.5}$ for each scenario. The numbers are average values, and numbers in parentheses denote one standard deviation (σ) from the average.**

| Scenario | Date of Feb. and time periods | $N_{INP}$ (# L$^{-1}$) | $N_{CPC}$ (# cm$^{-3}$) | AF×10$^3$ (%) | $n_S$×10$^{-8}$ (# m$^{-2}$) | $N_{500}$ (# cm$^{-3}$) | $N_{1000}$ (# cm$^{-3}$) | $m_{BC}$ (μg m$^{-3}$) | $m_{ammo}$ (μg m$^{-3}$) | $PM_{2.5}$ (μg m$^{-3}$) | $PM_{10}$ (μg m$^{-3}$) | $PM_{10-2.5}$ (μg m$^{-3}$) |
|---|---|---|---|---|---|---|---|---|---|---|---|---|
| Dust event | 21 *12:00-18:00* | 112 (34) | 3364 (952) | 3.6 (1.2) | 90.0 (30.1) | 26 (2) | 0.1 (0.1) | 0.4 (0.1) | 0.6 (0.1) | 57 (15) | 194 (26) | 137 (14) |
| Clean | 10, 17-18, 21-22 *11:00-23:00* | 4 (3) | 5205 (1557) | 0.1 (0.1) | 2.7 (4.4) | 67 (72) | 0.1 (0.1) | 1.1 (1.0) | 4.9 (5.9) | 48 (36) | 74 (52) | 26 (34) |
| Car emission | *16:00-20:00* of clean days | 3 (3) | 4647 (1510) | 0.1 (0.1) | 1.0 (1.1) | 69 (31) | 0.2 (0.1) | 1.2 (0.4) | 4.2 (2.2) | 54 (13) | 96 (43) | 43 (40) |
| Truck emission | *21:00-23:00* of clean day | 4 (3) | 4671 (1522) | 0.1 (0.1) | 0.6 (0.5) | 191 (89) | 0.1 (0) | 2.7 (1.2) | 15.7 (7.8) | 108 (42) | 132 (20) | 24 (25) |
| Pollution | 12-13, 24-27 *8:00-23:00* | 6 (8) | 4310 (964) | 0.1 (0.2) | 0.6 (0.9) | 279 (111) | 0.1 (0.1) | 2.5 (1.1) | 9.7 (4.6) | 130 (41) | 160 (45) | 30 (9) |
| Firework emission | *8:00-22:00* of 26 | 2 (3) | 4474 (993) | 0.1 (0.1) | 0.2 (0.3) | 324 (40) | 0.1 (0) | 2.2 (0.4) | 10.0 (1.8) | 135 (12) | 168 (12) | 34 (9) |
| Overall | 10-28 *(8:00-23:00)* | 12 (28) | 4370 (1229) | 0.4 (0.9) | 6.7 (22.6) | 209 (141) | 0.1 (0.1) | 1.9 (1.2) | 7.5 (5.4) | 102 (53) | 138 (60) | 35 (34) |

In the following sections, the data collected in this study is categorized into aforementioned scenarios. The correlation between $N_{INP}$ and the physiochemical properties of ambient particles, including particle number concentration and chemical composition, in each scenario and the potential source of immersion INP are discussed. We also compare to $N_{INP}$ measured under similar conditions reported in the literature, with particular attention to the IN activity and INP source attribution.

## 3.2 Contribution of mineral dust to $N_{INP}$ during the dust event

On the afternoon of Feb. $21^{st}$, 2021, a dust event occurred at the sampling site, as indicated clearly by the significant difference between $PM_{10}$ and $PM_{2.5}$ mass concentrations in Fig. 3c. $PM_{10}$ mass concentration reached 250 µg m$^{-3}$ and was 3 to 5 times as much as the $PM_{2.5}$ mass concentration during the dust event.

The dust event is characterized with substantially higher AF (0.0036% $\pm$ 0.0011%) and $n_S$ ($9.0\times10^9$ $\pm 3.0\times10^9$ # m$^{-2}$) compared to other days, as listed in Table 1. During the dust event, $N_{INP}$ was 1 to 2 orders of magnitude higher than clean days, ranging from 40 to 160 # L$^{-1}$. Meanwhile, the ambient particle number concentration entering HINC ($N_{CPC}$) during the dust event is only half to two thirds of the clean-day concentration level (Table 1), leading to the distinguishably higher AF and $n_S$. The significant increase of $N_{INP}$ in the dust event complies with the results of Bi et al. (2019), who reported $N_{INP}$ as high as 2800 # L$^{-1}$ measured at -30 ℃ and $RH_w$ = 106.5% during a desert dust event at a rural sampling site in suburban Beijing. $n_S$ during the dust event is also on the same order of magnitude with Bi et al. (2019) and Lacher et al. (2018), as shown in Fig. 5. However, most of the parameterizations obtained from laboratory experiments using Asian dust (AD) samples tend to overestimate $n_S$ by 1-2 orders of magnitude (Connolly et al., 2009; Niemand et al., 2012; Ullrich et al., 2017), except for the size-segregated parameterization proposed by Reicher et al. (2019) that spans from $1.9\times10^9$ # m$^{-2}$ for sub-micron to $4.2\times10^{10}$ # m$^{-2}$ for super-micron particles. 24-hour back trajectory analysis at three heights (20 m, 500 m, and 1000 m, Fig. A3) using National Oceanic and Atmospheric Administration (NOAA) HYSPLIT model suggests that the air parcel during the dust event is from the northwestern direction of Beijing, originating from the Mongolia Gobi Desert. The back trajectory analysis by Bi et al. (2019) also suggested that air parcels from Mongolia Gobi Desert tended to carry loads of desert dust, leading to higher $N_{INP}$.

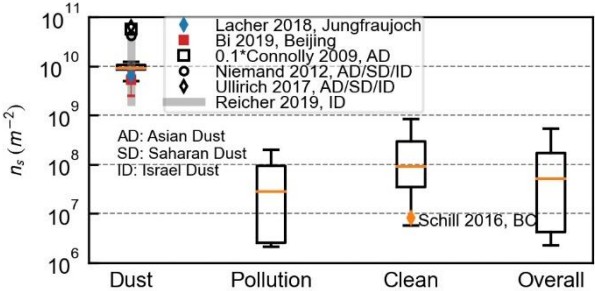

**Figure 5: Ice-active surface site density ($n_S$) for different scenarios. The median (horizontal orange lines), 25$^{th}$ and 75$^{th}$ percentiles (lower and upper boxes), and 10$^{th}$ and 90$^{th}$ percentiles (lower and upper whiskers) are shown.**

Previous laboratory studies have shown that larger particles, especially those larger than 500 or 1000 nm, exhibit superior
INP activity based on surface active site density theory (e.g., Connolly et al., 2009; Welti et al., 2009; Lüönd et al., 2010;
Hoose and Möhler, 2012; Ardon-Dryer and Levin, 2014; Chen et al., 2021). Bi et al. (2019) observed significant increase of
$N_{INP}$ when ambient particle peak size shifted towards the larger end of size spectra (exceeding 1000 nm) during
springtime dust events in rural Beijing. $N_{500}$ is generally below 50 # $cm^{-3}$ during springtime dust events in Beijing (Bi et al.,
2019), and is comparable to the results presented in Fig. 6a. To quantify the impact of $N_{500}$ on $N_{INP}$, linear regression analysis
between logarithms of $N_{INP}$ and $N_{500}$ is performed for dust event using ordinary least square (OLS) method, as shown in Fig.
6a. The triangles denote measured data, and the blue markers are the predicted data using the fitted linear regression parameters.
The Pearson's correlation coefficient ($r$) between $\log_{10}(N_{INP})$ and $\log_{10}(N_{500})$ is -0.3 (Fig. 6a), reflecting that $N_{500}$ is slightly
correlated with $N_{INP}$, and might have limited impact on $N_{INP}$ during the dust event. Besides, the significant difference between
$PM_{10}$ and $PM_{2.5}$ mass concentrations (Fig. 3c and Table 1) implies that large (dust) particle (occupies high mass concentration
but low number concentration) with high IN activity appeared during the dust event. An earlier study in east Mediterranean
urban region claimed that immersion IN activity of particles collected during dust storms correlated well ($R^2 = 0.47$) with
$PM_{10-2.5}$ between -10 ℃ and -30 ℃ (Ardon-Dryer and Levin, 2014). To explore the connection between $PM_{10-2.5}$ and $N_{INP}$ in
the urban environment, correlation analysis between $N_{INP}$ and $PM_{10-2.5}$ data collected during the dust event is also conducted
using OLS linear regression. The correlation between $N_{INP}$ and $PM_{10-2.5}$ at -30 ℃ ($r = -0.5$, Fig. C1) during the dust event in
this study suggests that $PM_{10-2.5}$ has a moderate negative correlation with $N_{INP}$ in the urban environment, which is stronger yet
not statistically significant. As suggested by previous studies (e.g., Atkinson et al., 2013; Kaufmann et al., 2016; Iwata and
Matsuki, 2018), dust mineralogy might be a superior immersion IN driving factor instead of $N_{500}$ and $PM_{10-2.5}$ in the urban
environment and worth further exploration.

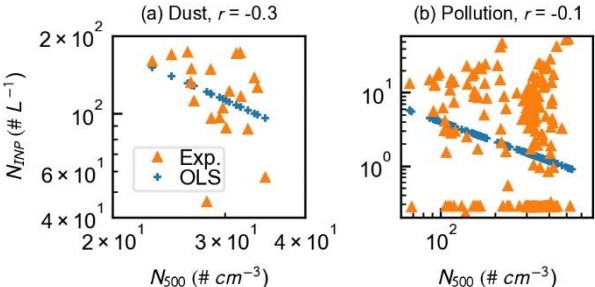

**Figure 6: Correlations between $N_{500}$ and $N_{INP}$ for (a) dust event and (b) heavy pollution. $r$ above each panel is the Pearson's**
**correlation coefficient for the linear regression fitting of the experiment data using ordinary least square (OLS) method. Blue**
**markers indicate predicted $N_{INP}$ using fitted linear regression parameters.**

The upper panel of Fig. 7 displays the diurnal profile of measured $N_{INP}$ and mass concentration of atmospheric ammonium
($m_{ammo}$) during the dust event. It can be seen that except for 13:00 (UTC+8), $N_{INP}$ profile seems to follow $m_{ammo}$ profile. Wu et
al. (2020) reported that ammonium ions could form and accumulate on mineral dust surface in the form of ammonium nitrate

in the highly populated urban environment. Previous laboratory studies have shown that the ammonium content on mineral dust surface might promote their IN activity due to strengthened ammonium ion surface adsorption followed by the formation of ice-favorable structure on dust particle surfaces (Boose et al., 2016b; Kumar et al., 2018; Whale et al., 2018; Kumar et al., 2019). To quantify the correlation between atmospheric ammonium content and $N_{INP}$ during the dust event, and to investigate

whether the observed enhancement of mineral dust IN activity by ammonium salts in previous studies (Boose et al., 2016b; Kumar et al., 2018; Whale et al., 2018; Kumar et al., 2019) still holds for the urban environment, linear regression analysis between $N_{INP}$ and $m_{ammo}$ is performed, as shown in the lower panel of Fig. 7. The blue markers are fitted $N_{INP}$ based on the OLS regression parameters, and the blue shading refers to $\pm 1\sigma$ range (calculated from measured $N_{INP}$) from fitted $N_{INP}$. $N_{INP}$ exhibits a moderate positive correlation with $m_{ammo}$ ($r = 0.5$), with more than 60% of measured $N_{INP}$ fall into the shaded area, suggesting

that $N_{INP}$ might be associated with $m_{ammo}$ during dust events in the urban environment. It should be noted that $N_{INP}$ also has a weak positive correlation with $m_{SO4}$ ($r = 0.4$, Table C1) during the dust event, but previous studies have confirmed that ammonium content, instead of anion species, is more likely to be the driving force in altering the immersion IN activity of mineral dusts (e.g., Kumar et al., 2018; Whale et al., 2018; Kumar et al., 2019). More field observations in urban areas, as well as systematic laboratory studies using natural mineral dust samples (e.g., Saharan dust and Asian dust, etc.) are required to

further investigate the connection between mineral dust surface characteristics and IN activity, and the underlying mechanism.

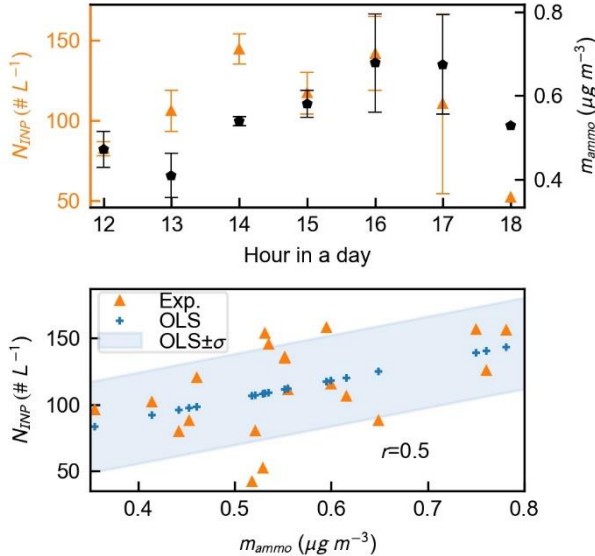

**Figure 7: Diurnal profile of (upper panel) and correlation between (lower panel) $N_{INP}$ and $m_{ammo}$ during the dust event. The blue markers in the lower panel indicate the predicted $N_{INP}$ using fitted linear regression parameters. The blue shading indicates $\pm 1\sigma$ of measured $N_{INP}$ from predicted $N_{INP}$.**

$N_{INP}$ measured during the dust event are compared with previous empirical parameterizations with a particular focus on mineral dusts to link parameterizations with the observation and gain better insight into the deviation between measured and predicted $N_{INP}$. Almost all measured $N_{INP}$ lie within a factor of 2.5 of the predicted $N_{INP}$ using the widely-used parameterization

proposed by DeMott et al. (2010) based on 14-year field observation data, as shown by the brown squares and shading in Fig. 8. This parameterization links $N_{INP}$ at a specific temperature with the exponential form of $N_{500}$ and takes not just mineral dusts but all ambient IN active aerosol types into consideration. It should be noted that most measured $N_{INP}$ lie above the DeMott et al. (2010) parameterization predicted $N_{INP}$. Such underestimation by DeMott et al. (2010) parameterization for ammonium-rich mineral dusts was also reported earlier in a Saharan dust plume observation, and was attributed to not taking the IN activity enhancement of mineral dusts by trace amount atmospheric ammonium into consideration (Boose et al., 2016b). Another empirical $N_{INP}$ parameterization proposed by DeMott et al. (2015) is specifically for mineral dusts based on laboratory measurement. However, it tends to systematically overestimate $N_{INP}$ during the dust event by up to an order of magnitude (3 to 13 times higher than the measured values) as shown by the blue squares in Fig. 8. Such overestimation on $N_{INP}$ during the dust event by previous parameterizations might be caused by omission of mineral dust chemical composition and mixing state change during transportation (e.g., Tobo et al., 2010; Wang et al., 2014; Li et al., 2016; Tang et al., 2016; Wu et al., 2020). Previous laboratory experiments suggest that even though mixing with organics might not affect the immersion IN activity of mineral dusts (Tobo et al., 2012; Wex et al., 2014; Kanji et al., 2019), mixing with sulfuric acids (Cziczo et al., 2009; Eastwood et al., 2009; Chernoff and Bertram, 2010; Niedermeier et al., 2011; Tobo et al., 2012; Augustin-Baudditz et al., 2014) and ammonium and sulfate salts (Cziczo et al., 2009; Iwata and Matsuki, 2018; Kumar et al., 2018; Whale et al., 2018; Kumar et al., 2019) could suppress the immersion IN activity of mineral dusts to different degrees. However, current parameterizations could represent the upper limit of atmospheric INP number concentration in global models, and we suggest that future parameterizations should include the influence of atmospheric processes (such as photo-oxidation and gaseous species condensation) on mineral dust IN activity to achieve more realistic prediction.

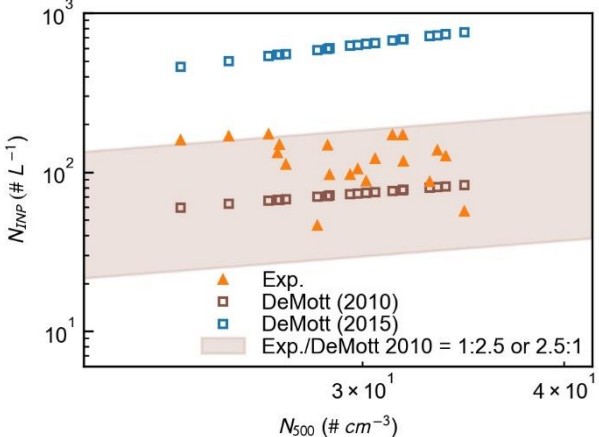

**Figure 8: Inter-comparison of measured (solid orange triangles) and predicted (hollow blue and brown squares) $N_{INP}$ during the dust event. The prediction is based on the parameterizations that link $N_{INP}$ with the number concentration of particles larger than 500 nm ($N_{500}$) at a specific temperature (DeMott et al., 2010; DeMott et al., 2015).**

### 3.3 Contribution of black carbon (BC) to $N_{INP}$

#### 3.3.1 Traffic emission

Clean days, when compared with heavy polluted or dusty days, provide ideal background to investigate the impact of primary particle emission sources, especially local traffic emission (from clean days), on $N_{INP}$ in urban regions. The major particle formation pathway in gasoline passenger vehicle exhausts is volatile organic compound (VOC) nucleation, producing large number of nanoparticles (diameter smaller than 50 nm) with low mass concentration (Raza et al., 2018 and references therein). On the other hand, diesel engine particle emission is dominated by BC particles ranging between 80-200 nm (Kittelson, 1998). The increase of mass concentrations of organics ($m_{org}$) and BC between 16:00 to 20:00 (UTC+8) in Fig. 9 corresponds to the evening rush hours, during which the emission of gasoline passenger vehicles dominating ambient particle population in the urban region. There is a further increase of $m_{Org}$ and $m_{BC}$ after 20:00 (UTC+8) followed by a plateau in Fig. 9. According to Beijing municipal administrative regulation, heavy-duty diesel trucks for goods transportation, as well as gasoline passenger vehicles with foreign plates (issued by cities other than Beijing) are only permitted to enter urban Beijing after 20:00 (UTC+8). Increasing emission from on-road heavy-duty diesel trucks and gasoline passenger vehicles with foreign plates are highly likely to be responsible for the increasing $m_{Org}$ and $m_{BC}$ after 20:00 (UTC+8; e.g., Hua et al., 2018; Zhang et al., 2019).

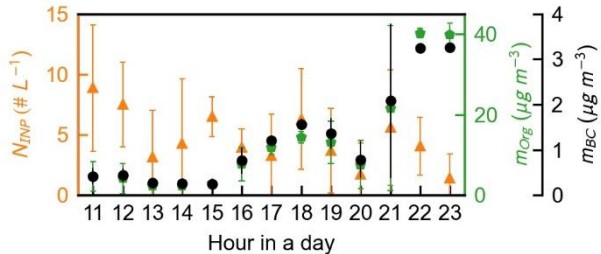

**Figure 9: Mass concentration of organics ($m_{org}$) and BC ($m_{BC}$) on clean days.**

$N_{INP}$ of clean days ranges between 0.3 to 16 # $L^{-1}$ which is the same order of magnitude as the immersion $N_{INP}$ results of Schill et al. (2016), who reported $N_{INP}$ for both freshly-emitted and aged BC on the orders of $10^{-1}$ to $10^1$ # $L^{-1}$ measured at similar experiment condition (-30 ℃ and $RH_w$ = 105%) to this study, using BC generated from an off-road diesel engine. AF and $n_S$ reported by Schill et al. (2016) lies in the lower range of measured AF and $n_S$ associated with vehicle emission periods (Table 1 and Fig. 5). To investigate the impact of traffic emission on $N_{INP}$, linear regression analysis is performed between $N_{INP}$ and $m_{BC}$, a widely used cursor of traffic emission. As shown in Fig. 10a, the correlation between $N_{INP}$ and $m_{BC}$ is poor ($r$ = 0), implying that $N_{INP}$ is independent of $m_{BC}$ on clean days. Recently, Kanji et al. (2020) also reported that BC might not act as effective immersion INP based on laboratory experiments. The absence of a relationship between $N_{INP}$ on $m_{BC}$ on clean days is consistent with previous findings (Schill et al., 2016; Kanji et al., 2020; Schill et al., 2020), suggests that BC associated with vehicle emission might not act as active immersion INP in the urban atmosphere.

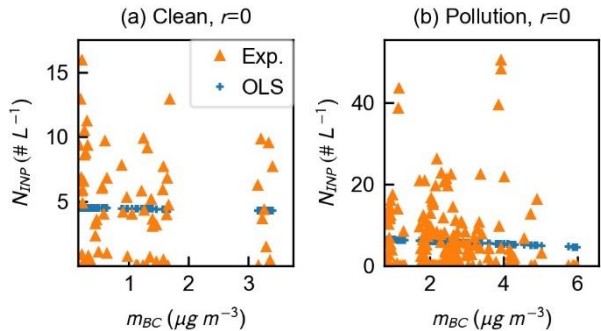

**Figure 10: Correlations between $m_{BC}$ and $N_{INP}$ during (a) clean and (b) heavy pollution periods. $R^2$above each panel is Pearson's correlation coefficient. Blue markers indicate the predicted $N_{INP}$ using fitted linear regression parameters.**

### 3.3.2 Firework emission

Heavy pollution accompanied by the presence of substantially higher mass concentrations of non-refractory $PM_1$, $PM_{2.5}$ (128 $\pm 44$ µg m$^{-3}$) and BC (up to 6 µg m$^{-3}$) occurred during Feb. 12-13 and 24-27, as shown in Fig. 2-4. $N_{500}$ was also substantially higher during heavy pollution than other days (Table 1). As stated above, large amounts of particles comprising carbonaceous material and chloride emerged after Spring and Lantern Festival celebrations due to firework emissions which also contain trace amounts of potassium and other metal elements (Jiang et al., 2015; Kong et al., 2015; Cao et al., 2017).

With almost 10 times as much $N_{500}$ as that of the dust event during heavy pollution, it was expected that a substantially higher INP concentration should be observed if these large particles are effective INP. However, the particle population during heavy pollution did not exhibit superior IN activity, with the majority of $N_{INP}$ fell into the range of 0 to 25 # L$^{-1}$. $\log_{10}(N_{INP})$ exhibits a weak negative correlation with $\log_{10}(N_{500})$ in Fig. 6b ($r = -0.2$). As shown in Fig. 10b, the OLS linear regression results further suggest that $N_{INP}$ is likely to be independent of $m_{BC}$ during heavily polluted days ($r = 0$). The independence of $N_{INP}$ on $m_{BC}$ is compliant with the results of Adams et al. (2020), in which there was a substantial growth (by more than an order of magnitude) of ambient particle number concentration and $m_{BC}$ from combustion and firework emissions, but no significant $N_{INP}$ change was observed. Chen et al. (2018) conducted off-line $N_{INP}$ measurement using filtered samples collected at the same sampling site as this study during heavy pollution, and found no dependence of $N_{INP}$ on neither the mass concentrations of $PM_{2.5}$ and BC, nor $N_{500}$ as well. As aforementioned, the synergetic heavy pollution after festival celebrations were induced by secondary pollutant formation via liquid phase reaction (e.g., Wang et al., 2018; Wu et al., 2018) and firework emission. Under the mixed-phase cloud condition (-30 °C, $RH_w = 104\%$) in this study, such particles are very likely to become aqueous droplets or contain liquid films on solid particles, which might require conditions for homogeneous freezing to nucleate ice.

## 4. Conclusion

In situ observation of $N_{INP}$ and physiochemical properties, including chemical composition and size distribution, of ambient particles at an urban site in Beijing during the traditional Chinese Spring Festival has been performed at mixed-phase cloud condition (-30 °C, $RH_w$ = 104%) for 18 days. The impact of different scenarios, such as the synergetic heavy pollution induced by secondary aerosol formation and firework emissions, a dust event, and local traffic emissions on $N_{INP}$ has been explored. $N_{INP}$ was investigated in relation to $N_{500}$ and $m_{BC}$. The relationships of $m_{ammo}$ with $N_{INP}$, as well as $PM_{10-2.5}$ with $N_{INP}$, during the dust event are also presented. The results show that $N_{INP}$, as well as AF of ambient particles during dust event are substantially higher than all other scenarios. $N_{INP}$ could reach 160 # $L^{-1}$ during the dust event, while it ranges from $10^{-1}$ to $10^{1}$ # $L^{-1}$ on other days. AF and $n_S$ during the dust event (0.0036% ±0.0011% and $9.0 \times 10^9$ ±$3.0 \times 10^9$ # $m^{-2}$) is 20 to 30 times higher than clean (0.0001% ±0.0001% and $2.7 \times 10^8$ ±$4.4 \times 10^8$ # $m^{-2}$) and heavily polluted days (0.0002% ±0.0002% and $6.5 \times 10^7$ ± $9.3 \times 10^7$ # $m^{-2}$). During the dust event, $N_{INP}$ exhibit a moderate positive correlation with $m_{ammo}$ ($r$ = 0.5) and a moderate negative correlation with $PM_{10-2.5}$ ($r$ = -0.5). The parameterization proposed by DeMott et al. (2010) predicts more than 60% of measured $N_{INP}$ within a factor of 2.5 during the dust event. Mass concentration measurements suggest that large amounts of aerosols containing chloride and BC appeared after the celebrations on Feb. 11[th] and 26[th] nights due to firework emission. Meanwhile, the stagnant and humid meteorology condition provides ideal condition for secondary aerosol formation. But there is no significant difference between $N_{INP}$ on heavily polluted and clean days, implying the urban aerosols from multiple sources with complex chemistry might not be effective INP. Besides, the diurnal increase of $m_{BC}$ from petrol passenger vehicle emissions during rush hours and from diesel truck emissions after 20:00 (UTC+8) on clean days does not lead to distinguishable higher $N_{INP}$, implying that local traffic emission also has negligible impact on $N_{INP}$. Our study reveals that immersion INP population in the urban environment has increased substantially during the East Asian dust event. Furthermore, our results agree with previous literature from laboratory and field studies that atmospheric BC from both local traffic and firework emissions has negligible effects on mixed-phase cloud formation, and that $N_{INP}$ is unaffected by heavy pollution.

**Appendix A Ambient aerosol characterization and back trajectory analysis of the dust event**

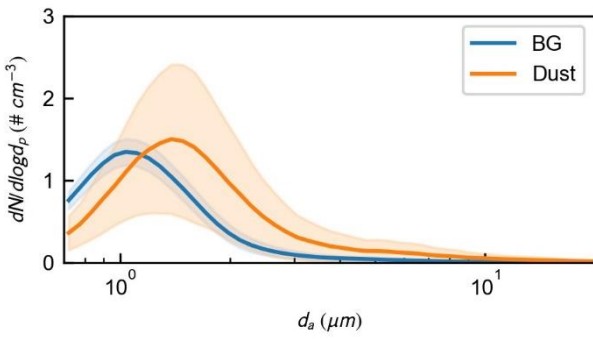

Figure A1. Aerodynamic diameter ($d_a$) size distribution of particles before (00:00-06:00, Feb. 21st; BG) and during the dust event (11:00-18:00, Feb. 21st; Dust).

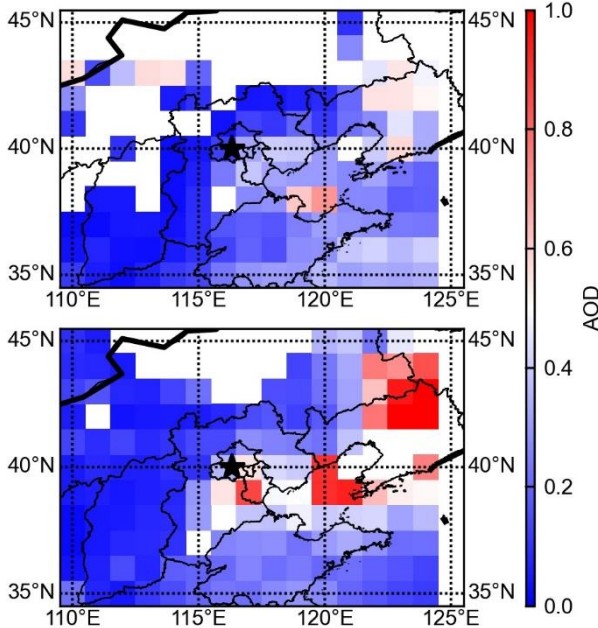

Figure A2. Aerosol optical depth (AOD) on Feb. 20th (upper panel) and Feb. 21st (lower panel) using MODIS Aqua Deep Blue Collection 6 aerosol data at 550 nm (Acker and Leptoukh, 2007). Sampling site is denoted with black star in each panel.

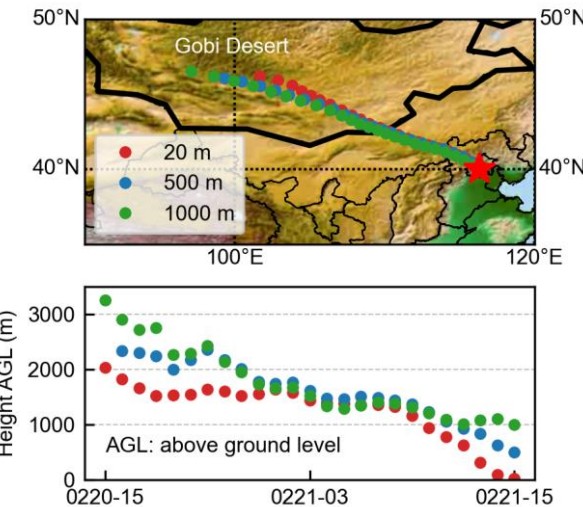

**Figure A3. 24-hour back trajectory of the air parcel at the sampling site at three different heights (20 m, 500 m, and 1000 m above ground level).**

## Appendix B HINC lamina condition calibration

HINC lamina condition calibration was performed by injection of 200 nm ammonium nitrate aqueous droplets into HINC at four lamina temperatures, i.e., -45 ℃, -40 ℃, -35 ℃, and -30 ℃. HINC was operated in *RH* scan mode during the calibration, in which HINC lamina temperature was maintained constant by varying the wall temperatures simultaneously, resulting in a temperature gradient and supersaturation in the chamber (Lacher et al., 2017). The $RH_i$ within HINC lamina changed continuously from 100% to 160% at each lamina temperature. The concentration of the ammonium nitrate solution was 0.0025 mol L$^{-1}$. The solution was atomized by a nebulizer (TSI Inc.) using 1.5 LPM nitrogen gas. The flow stream was dried to $RH_w$ < 2% by passing through a 47 cm Nafion$^{TM}$ dryer and was then size selected by a DMA (model 3081 long; TSI Inc.). Particle number concentration entering HINC was measured online by a CPC (model 3775; TSI Inc.).

Figure B1 shows the AF as a function of $RH_w$ at -30 ℃ for 200 nm ammonium nitrate aqueous droplets. Different OPC channel (> 1 μm, > 2 μm, > 3 μm, > 5 μm) are marked with different colors. It is worth noting that given the flow structure of HINC in this study (Sect. 2.2.3), only particles larger than 5 μm detected by HINC OPC would be recognized as detectable ice crystals (below -38 ℃) or water droplets (above -38 ℃). As shown in Fig. B1, 200 nm ammonium nitrate aqueous droplets start to grow upon water saturation (black dots), followed by more rapid growth with increasing $RH_w$ (grey dots). However, there is no growth in >5 μm channel until $RH_w$ exceeds 106%, corresponding to the presence of detectable water droplets larger than 5 μm. Therefore, HINC should be operated below 106% at -30 ℃ to avoid erroneous count of large (>5 μm) water droplets rather than ice crystals.

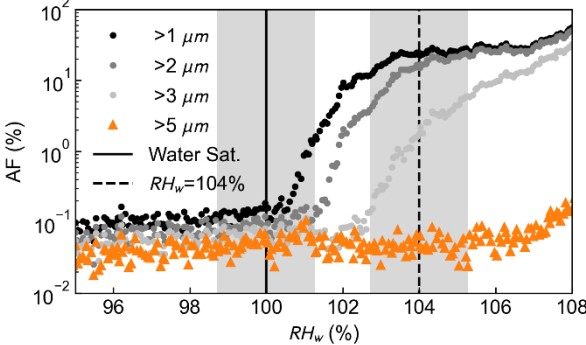

**Figure B1. AF as a function of $RH_w$ at -30 ℃ for 200 nm ammonium nitrate aqueous droplets detected in different HINC OPC channels. Vertical solid and dash black lines represent water saturation ($RH_w$ = 100%) and $RH_w$ = 104%, respectively. Grey shading indicates the average variation of $RH_w$ along HINC lamina centerline.**

Figure B2 shows the IN onset or water droplet survival points for 200 nm ammonium nitrate during the calibration. The IN onset or water droplet survival points are defined as the temperature and *RH* when 0.1% of aerosols entering HINC are activated into detectable ice crystals or water droplets by HINC OPC in >5 μm channel. The blue and black solid lines represent water saturation ($RH_w$ = 100%) and homogeneous freezing threshold (Koop et al., 2000), respectively. The error bars in Fig. B2 represent one standard deviation of temperature and $RH_i$ along HINC lamina centerline for each individual $RH_i$ scan. During

the calibration, the average variation of lamina $RH_w$ was less than 1.2% (corresponding to a 1.8% variance of lamina $RH_i$). Lamina $RH_w$ suffers larger variation as $RH$ increases, resulting in $RH_w = 108\% \pm 2.1\%$ at -30 ℃. The variance of lamina

temperature was below 0.2 K throughout the calibration process. As shown in Fig. B1, the IN onset point of 200 nm ammonium nitrate at -40 ℃ lies on the calculated homogeneous freezing threshold. The IN onset at -45 ℃ exceeds the homogeneous freezing threshold by 3.5%, yet still below water saturation line. When the lamina temperature is above -38 ℃, water drops require $RH_w$ substantially higher than 104% (dashed line) to be detected in the >5 μm OPC channel, as such we are confident that signals arising in the >5 μm OPC channel at $RH_w = 104\%$ are due to ice crystal formation.

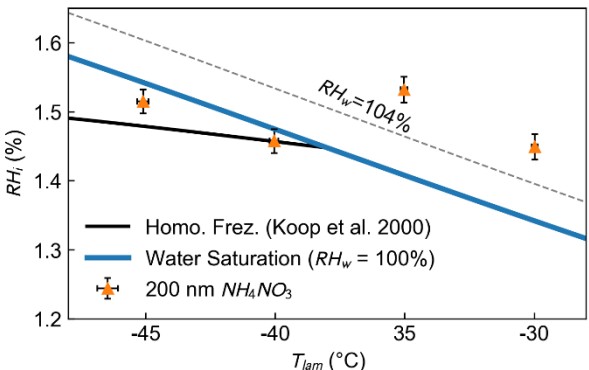

**Figure B2. Phase diagram of the IN onset (below -38 ℃) or water droplet detection in the >5 μm OPC channel (above -38 ℃) for freezing or water droplet formation onto 200 nm dry diameter ammonium nitrate particles. The solid blue and black lines represent the water saturation line and the homogeneous freezing threshold of 200 nm aqueous droplets (Koop et al., 2000), respectively. The horizontal and vertical error bars represent the variation of temperature and $RH_i$ along HINC lamina centerline.**

# Appendix C Supplemental Information

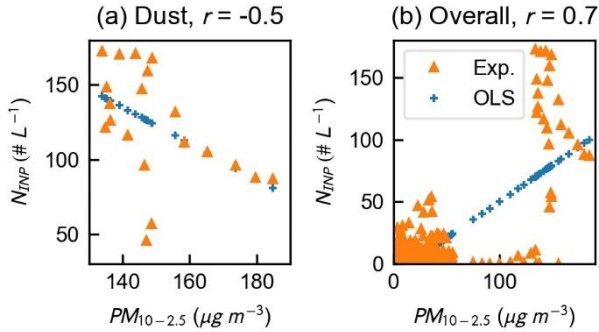

**Figure C1: Correlations between $PM_{10-2.5}$ and $N_{INP}$ in the (a) dust event and (b) observation. *r* above each panel is Pearson's correlation coefficient. Blue markers indicate the predicted $N_{INP}$ using fitted linear regression parameters.**

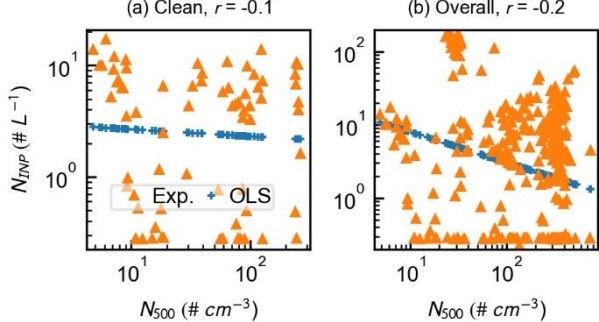

**Figure C2: Correlations between $N_{500}$ and $N_{INP}$ for (a) clean days and (b) the observation. *r* above each panel is Pearson's correlation coefficient. Blue markers indicate the predicted $N_{INP}$ using fitted linear regression parameters.**

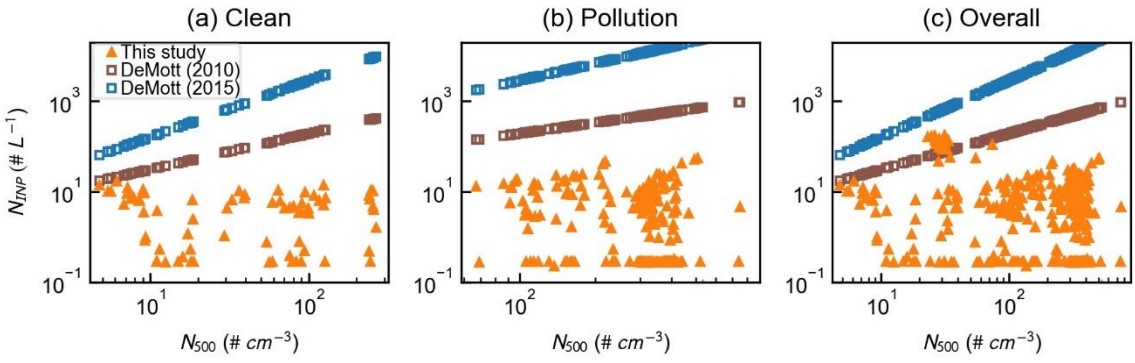

**Figure C3: Inter-comparison of measured (solid orange triangles) and predicted (hollow blue and brown squares) $N_{INP}$ on clean (upper panel), heavy pollution (middle panel) days, and during the observation (lower panel). The prediction is based on the parameterizations that link $N_{INP}$ with the number concentration of particles larger than 500 nm ($N_{500}$) at a specific temperature (DeMott et al., 2010; DeMott et al., 2015).**

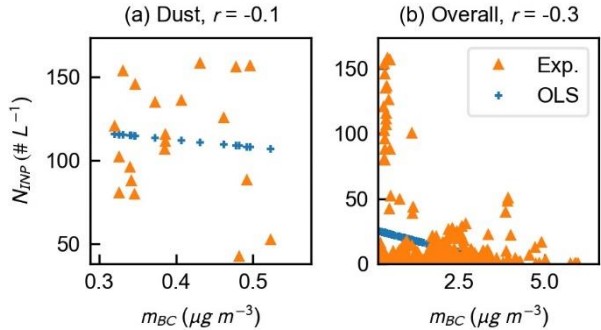

**Figure C4: Correlations between $m_{BC}$ and $N_{INP}$ during the (a) dust event and (b) observation. $r$ above each panel is Pearson's correlation coefficient. Blue markers indicate the predicted $N_{INP}$ using fitted linear regression parameters.**

**Tabel C 1 Pearson's correlation coefficient ® between $N_{INP}$ and selected measured values.**

| Measured Values | $N_{500}$ | $N_{1000}$ | $m_{Org}$ | $m_{NH4}$ | $m_{SO4}$ | $m_{NO3}$ | $m_{BC}$ | $PM_1$ | $PM_{2.5}$ | $PM_{10}$ | $PM_{10\text{-}2.5}$ |
|---|---|---|---|---|---|---|---|---|---|---|---|
| Dust Event | -0.36 | -0.24 | -0.04 | 0.46 | 0.36 | 0.06 | -0.09 | 0.06 | 0.05 | -0.23 | -0.49 |
| Overall | -0.29 | 0.01 | -0.32 | -0.32 | -0.34 | -0.37 | -0.30 | -0.37 | -0.19 | 0.22 | 0.70 |

**Data availability**

Data inquires can be directed to the corresponding author (Zhijun Wu, zhijunwu@pku.edu.cn).

**Author Contributions**

CZ and ZW designed the experiments and methodology. CZ and JCC conducted the field observation. CZ, WZ, and LT performed aerosol chemical and size distribution analysis. CZ, JC, XP,  and JCC calibrated HINC. SC and LZ provided the meteorology data. MH, ZW, ZAK, PT, and SG supervised the observation. CZ, ZW, and ZAK prepared the manuscript with input from all coauthors.

**Competing interests**

The authors declare no competing interests.

**Acknowledgments**

This work was supported by National Natural Science Foundation of China (NSFC, grant no. 42011530121, 91844301) and Ministry of Science and Technology of China (MOST, grant no. 2019YFC0214701). Aerosol optical depth data used in this

paper were produced with the Giovanni online data system, developed and maintained by the NASA GES DISC. ZAK acknowledges funding from the Atmospheric Physics Chair, ETH Zurich.

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
