# Peer review of "Ice nucleating particles from multiple aerosol sources in the urban environment of Beijing under mixed-phase cloud conditions"

_Atmospheric Chemistry and Physics, 2021_

## Author Response (AR1)

Dear editor,

We would like to thank the referees for their valuable comments that help us to substantially improve the quality of this manuscript. We have made every effort to address referees' comments with associated revisions to the manuscript. Please see below our point-by-point response to referee comments.

For clarity, referee comments are shown in *italic font*; the response to the comments are shown in blue font, the notes and explanations for the changes in the manuscript are shown in underline blue font; the revised contents in the manuscript are shown in green font. Our line numbers refer to the updated manuscript, not the original one. Detailed corrections can be found in the marked-up manuscript file after our response.

**Referee #1**

**Summary:**

*The study presented by C. Zhang and co-authors investigates the variability of immersion-mode INP concentrations at -30 °C in an urban environment. INPs were quantified with a high-time resolution which allows to classify daily INP variability, and to investigate the impact of multiple aerosol sources such as fireworks during festivities, local traffic, secondary aerosol particle formation and mineral dust. While only during dust impacted times the ice nucleation ability of the aerosol population was found to be higher as the background, no dependence on aerosol particle concentration larger than 500 nm is observed. Still, the majority of INP observations are predictable within a factor of 2.5 using the parameterization proposed by DeMott et al. (2010) during the dust event. In general, no effect from anthropogenic activities on INP number concentration is observed. The study will be of interest for the ice nucleation community, as it strengthens the findings that anthropogenic aerosol particles are not contributing to the ice formation in mixed-phase clouds. The manuscript is very well written and I only have minor comments.*

We thank the referee for the recognition of our work. Below, the referee outlines concerns on the number concentration to be used in correlation analyses and particle transmission efficiency of the sampling inlet. Besides, the referee suggests to highlight the novelty of this work and expand the background information. We have made necessary revisions which we believe have sufficiently addressed the referee's concerns.

**Minor comments:**

1. *I recommend to highlight better the novelty of this study over existing ones. E.g., Chen used filters to quantify INP concentrations at temperatures above -25 °C; Chen et al. (2019; 2022) and Bi et al. (2019) did not compare their measurements to an AMS, which allows to distinguish different anthropogenic aerosol emission sources.*

[Response]: We thank the referee for the suggestion. Added a few lines in the introduction between L92-L93 and L101-L102 to better highlight the novelty of this study.

L92-L93: "The lack of online particle chemistry information impedes aerosol source correlation in these studies (Chen et al., 2018; Bi et al., 2019)."

L101-L102: "Urban particle emission sources were distinguished based on the online chemical characterization using an Aerosol Chemical Speciation Monitor (ACSM)."

2. *The correlation coefficient analysis of the INP concentration measurements using HINC and aerosol particle size distribution does not consider the cut-off size of HINC. Therefore, I believe that the results can be biased by the non-sampling of larger particles in HINC. I recommend to first define the cut-off size of HINC, and only compare the size distributions measurements in this size range with the INP concentration measurements.*

[Response]:  Thank you for pointing it out, this is a good point. The gravitational settling for 1 µm, 2 µm, and 5 µm particles in the tubing between APS+SMPS and HINC inlet are estimated to be 2.1%, 7.7%, and 41.0%, respectively (Brockmann, 2011, eq. 6-51 to 6-53). Such particle loss is considered to be negligible because particles larger than 1 µm are scarce with $N_{1000}$ always below 1 # cm$^{-1}$ during the observation.

The particle loss in HINC chamber in this study is consistent with Lacher et al. (2017), i.e. 23.5% for 1 µm particle, 46.6% for 1.5 µm particles, and 100% for particles larger than 5 µm (Brockmann, 2011, eq. 6-51 to 6-53). Therefore, we define the $d_a$ at which 50% particle could penetrate HINC to be $d_{50}$ = 1.5 µm.

Figure R1-1 shows the correlation analysis using $N_{500}$ calculated with (orange triangles) and without (grey triangles) taking HINC $d_{50}$ = 1.5 µm into consideration for different scenarios. The results are unaffected by the number concentration of particles larger than1.5 µm in all scenarios, because particles larger than 1.5 µm were scarce as stated above.

Added "Estimated particle gravitational settling within HINC is consistent with the OPC measurement by Lacher et al. (2017), i.e. ~23.5% for 1 µm particle, 46.6% for 1.5 µm particles, and 100% for particles larger than 5 µm (Brockmann, 2011, eq. 6-51 to 6-53). Therefore, only particles larger than 5 µm detected by the HINC OPC are counted as ice crystals. The INP concentration measurements in this study are representative of ambient particles smaller than 1.5 µm. For more detailed HINC design and operating principle information, please refer to Lacher et al. (2017) and Kanji and Abbatt (2009). In addition, the gravitational settling estimated for 1 µm, 2 µm, and 5 µm particles in the tubing upstream HINC inlet are estimated to be ~2.1%, 7.7%, and 41.0%, respectively (Brockmann, 2011, eq. 6-51 to 6-53). Such sampling line loss is considered to be negligible because particles larger than 1 µm were scarce during the observation (Table 1)." between L147-L155.

The notations $N_{500}$ and $N_{1000}$ are explicitly defined as "number concentration of particles ranging from 500 nm to 1.5 µm ($N_{500}$) and 1000 nm to 1.5 µm ($N_{1000}$)" in L220.

[Figure]

**Figure R1-1. Correlations between $N_{INP}$ and calculated $N_{500}$ with (orange) and without(grey) taking HINC cut-off size into consideration in the dust event.**

L147-L155: "Estimated particle gravitational settling within HINC is consistent with the OPC measurement by Lacher et al. (2017), i.e. ~23.5% for 1 μm particle, 46.6% for 1.5 μm particles, and 100% for particles larger than 5 μm (Brockmann, 2011, eq. 6-51 to 6-53). Therefore, only particles larger than 5 μm detected by the HINC OPC are counted as ice crystals. The INP concentration measurements in this study are representative of ambient particles smaller than 1.5 μm. For more detailed HINC design and operating principle information, please refer to Lacher et al. (2017) and Kanji and Abbatt (2009). In addition, the gravitational settling estimated for 1 μm, 2 μm, and 5 μm particles in the tubing upstream HINC inlet are estimated to be ~2.1%, 7.7%, and 41.0%, respectively (Brockmann, 2011, eq. 6-51 to 6-53). Such sampling line loss is considered to be negligible because particles larger than 1 μm were scarce during the observation (Table 1)."

L220: "…number concentration of particles ranging from 500 nm to 1.5 μm ($N_{500}$) and 1000 nm to 1.5 μm ($N_{1000}$)…"

3.  *Introduction: As you give an overview of the impact of anthropogenic aerosol particles on ice nucleation for heterogeneous freezing in the mixed-phase and cirrus cloud regime it might be useful for the reader to explain a bit more about these freezing modes and cloud regimes.*

[Response]: Thanks for the suggestion! We added a brief introduction on the freezing modes and clouds regimes between L34-L37.

L34-L37: "Ice crystals in clouds can form via homogeneous freezing of aqueous droplets below -38 ℃, or via heterogeneous ice nucleation (IN) with the aid of foreign interfaces offered by atmospheric ice nucleating particles (INPs) through immersion/contact freezing of existing droplets at higher temperature or direct deposition/condensation of water vapor below water saturation (Pruppacher and Klett, 1997; Vali et al., 2015; Kanji et al., 2017)."

*Specific comments:*

1. *Lines 45 – 47: This statements requires a reference.*

[Response]: Reference added. Added "(e.g., Sun et al., 2016)" after the statement.

L50: "Most of the particles … inorganic salts **(e.g., Sun et al., 2016)..**"

2. *Line 47: What do you precisely mean with "regional transport"? Regional transport of e.g., dust particles?*

[Response]: To be more clear, added "…, such as transportation of mineral dusts from deserts and pollutants from adjacent areas,…" after "regional transportation".

We used "regional transport" here to differentiate "nonlocal emission" from "local emission".

L50-52: "Apart from local emissions, regional transportation**, such as transportation of mineral dusts from deserts and pollutants from adjacent areas,** also contributes significantly to urban particle population under appropriate meteorological conditions…"

3. *Lines 49 – 50: These are examples for this statement, more literature exist about the impact of aging on aerosol properties. Please include "e.g." to the citations or complete the list of citations.*

[Response]: Thanks! Added "e.g., " before current citations.

L53-L54: "and mixing state (**e.g.,** Lin et al., 2016; Sun et al., 2016; Hua et al., 2018; Zhang et al., 2020b; Lei et al., 2021; Li et al., 2021)".

4. *Line 52: "… and catalyze ice crystal formation below 0 °C…" this is almost unnecessary information; it would be more interesting to give information about the onset temperature of freezing for each particle type.*

[Response]: Deleted "… and catalyze ice crystal formation below 0 °C". Added a brief introduction to the freezing onset temperature for mineral dusts and bioaerosols.

L54-L59: "Previous studies have confirmed that several kinds of atmospheric particles, including mineral dusts, carbonaceous particles, and biogenic species, can act as immersion INP (Murray et al., 2012; Kanji et al., 2017 and references therein). When present at atmospherically relevant amounts in droplets, mineral dusts mostly catalyze super-cooled aqueous droplet freezing below -15 ℃ (Hoose and Möhler, 2012 and references therein), while biological species, such as

pollen, fungal spores, and viruses, generally exhibit immersion IN activity below -5 ℃ and are fully activated below -10 ℃ to -20 ℃ (e.g., Chou, 2011; Conen et al., 2015; Kanji et al., 2017; Conen et al., 2022; Porter et al., 2022)."

5.  *Lines 53 – 54: I find this statement problematic as you mix INP number concentration from two different cloud regimes, mixed-phase and cirrus clouds. Moreover, studies exist where INPs are measured at warmer temperatures (e.g., -8 ℃; Conen et al., 2015).*

[Response]:  Thanks! We have removed statements about cirrus clouds and kept those relevant to MPC, which is the scope of this study.

L60-L61: "The reported atmospheric immersion INP number concentration ($N_{INP}$) were measured between -5 ℃ and -38 ℃ and were normally on the order of $10^{-2}$ to $10^3$ # $L^{-1}$ (e.g., Rogers et al., 1998; DeMott et al., 2010; Chen et al., 2018; Porter et al., 2022)."

6.  *Lines 81 – 82: "In other words" sounds not correct in this context.*

[Response]: Deleted "In other words".

L82-L88: Chen et al. (2018) quantified off-line immersion INP concentration using filter samples collected every 12 hours during a heavily polluted 2016 wintertime in Beijing. Even though a high level PM2.5 with complex chemical composition was sampled during the pollution period in the urban area, these aerosols did not act as superior INPs, and the highest INP concentration measured at -26 ℃ was below 10 # $L^{-1}$, similar to what was observed in remote regions such as the Swiss Alps (Boose et al., 2016a; Lacher et al., 2017). The INP concentration reported by Chen et al. (2018) was insensitive to particle number concentration and particle chemistry in an atmosphere that was dominated by anthropogenic emissions.

7.  *Lines 107 – 108: Was this upper size limit for the inlet quantified in other studies, or did you measure this size-cut? Please give more information how this cut-off size was derived.*

[Response]: We estimate the aerodynamic size-resolved inlet transmission efficiency based on the method described in Brockmann (2011, eq. 6-23 to 6-29). The inlet transmission efficiency is dominated by gravitational settling and inertial loss in this study, as the aspiration efficiency ranges between 1 to 1.1 and the diffusion loss is negligible.

The average wind speed during HINC measurement period (2.08 m $s^{-1}$) is used as ambient gas velocity $U_0$, and the sampling gas velocity $U$ is 1.05 m $s^{-1}$ at the inlet plane. Figure R1-2 shows the inlet transmission efficiency as a function of $d_a$, with $d_{10}$ and $d_{50}$ (the $d_a$ at which 10% and 50% particle could be transported to instruments through the inlet, respectively) being ~20 µm and 13.4 µm, respectively.

[Figure]

**Figure R1-2. Transmission efficiency of the sampling inlet as a function of $d_a$ (Brockmann, 2011).**

The $d_{10}$ and $d_{50}$ information has been added in L116.

L115-L118: "The tube inlet was arched so that it was bent facing downwards (see Fig. 1) to prevent water contamination, with $d_{10}$ and $d_{50}$ (the $d_a$ at which 10% and 50% particle could be transported to instruments through the inlet, respectively) being ~20 μm and 13.4 μm, respectively (Brockmann, 2011, eq. 6-23 to 6-29)."

8.  *Line 116: I believe that this is the upper limit to operate this kind of SMPS; towards the larger sizes, you are likely impacted by the effect of double charged particles; please give an error estimate here.*

[Response]: This is a good point. We used a 0.0508 cm impactor ($d_{50}$ = 735 nm) at the DMA inlet to remove large particles, including large multiply-charged particles. Theoretically, at least 50% particles larger than 735 nm should be removed with the impactor. Multiple charging correction was further applied to the reported size distribution to minimize the impact of multiple charging.

L125-L127: "An impactor was installed at the SMPS inlet to remove particles larger than 735 nm. Furthermore, multiple charging correction was applied when processing the sub-micron particle number size distribution."

9.  *Line 119: Did you change the value of effective density when you were impacted by pollution? And if not, what was the reason to choose a constant value?*

[Response]: The effective density of ambient aerosol during heavy pollution and clean days in Beijing was reported to be constant around 1.5 g cm$^{-3}$ (Qiao et al., 2018; An et al., 2019). Reference updated in the manuscript.

L129-L132: "The aerodynamic particle number size distribution obtained from APS could be converted to particle mobility size ($d_m$) distribution by assuming the effective density of ambient particles to be 1.5 g cm$^{-3}$ for clean and polluted days (Khlystov et al., 2004; Chen et al., 2018; Qiao et al., 2018; An et al., 2019)."

10. *Line 130: What are the errors associated with the lamina temperature and the relative humidity with respect to water in HINC?*

[Response]: The one standard error deviation of lamina temperature and $RH_w$ uncertainties are respectively 0.2 ℃ and 2.2%. Added the corresponding one standard errors in L142-L143.

L142-L143: "…with a lamina temperature ($T_{lam}$) of -30 ±0.2 ℃ and $RH_w = 104 ±2.2\%$ (equivalent to $RH_i = 140 ±3.0\%$, where the subscript i denotes ice)."

11. *Line 145: What was the sampling averaging time in HINC, e.g., 20 minutes as in Lacher et al. (2017)? And how did you treat INP concentration measurements below the detection limit?*

[Response]: As stated in L160-L161, the sampling averaging time in HINC of this study was 15 minutes ("every 15 minutes of measurement"), with 5-minute background tests before and after the sampling period.

As for the treatment to the $n_{INP}$ below the detection limit, we report the positive values as is and replace the negative values with the minimum quantifiable concentration (~0.26 # L$^{-1}$) in the revised manuscript. The treatment is detailed between L162-L167.

L162-L167: "Average ice crystal concentration (equivalent to $N_{INP}$) of the 15-minute measurement is calculated by firstly subtracting the average background particle counts from measurement counts, and secondly converting particle counts to number concentration using HINC sampling flowrate. The subtraction might produce negative $N_{INP}$ when the signal of OPC during the measurement is undistinguishable from background noise. Therefore, this study reports positive $N_{INP}$ as is, and replaces negative $N_{INP}$ with the minimum quantifiable concentration of OPC (0.26 # L$^{-1}$) following the method in Lacher et al. (2017)."

12. *Figures 2 and 3: It might be useful to also plot the INP concentration timeseries in these plots.*

[Response]: Added $n_{INP}$ in panel c of Fig. 3.

[Figure]

**Figure R1-3 Updated Figure 3.**

13. *Table 1 and related discussion in the text: It might be interesting to the reader to also receive these information about local traffic and firework emission (e.g., as a sub-category of 'pollution').*

[Response]: Thanks for the suggestion! Three rows (car emission, truck emission, and firework emission) have been added in Table 1 to provide information for specific time periods associated with local traffic and firework emission.

**Updated Table 1. The date, number concentrations of immersion INP ($N_{INP}$) and ambient particle ($N_{CPC}$), activation fraction (AF), ice-active surface site density ($n_S$), number concentrations of particles ranging from 500 nm to 1.5 µm ($N_{500}$) and 1000 nm to 1.5 µm ($N_{1000}$), as well as mass concentrations of BC particles ($m_{BC}$) and ammonium salt ($m_{ammo}$), PM$_{2.5}$ and PM$_{10}$, and PM$_{10-2.5}$ for each scenario. The numbers are average values, and numbers in parentheses denote one standard deviation (σ) from the average.**

| Scenario | Date of Feb. and time periods | $N_{INP}$ (# L$^{-1}$) | $N_{CPC}$ (# cm$^{-3}$) | AF×10$^3$ (%) | $n_S$×10$^{-8}$ (# m$^{-2}$) | $N_{500}$ (# cm$^{-3}$) | $N_{1000}$ (# cm$^{-3}$) | $m_{BC}$ (µg m$^{-3}$) | $m_{ammo}$ (µg m$^{-3}$) | PM$_{2.5}$ (µg m$^{-3}$) | PM$_{10}$ (µg m$^{-3}$) | PM$_{10-2.5}$ (µg m$^{-3}$) |
|---|---|---|---|---|---|---|---|---|---|---|---|---|
| Dust | 21 | 112 | 3364 | 3.6 | 90.0 | 26 | 0.1 | 0.4 | 0.6 | 57 | 194 | 137 |
| | *12:00-18:00* | (34) | (952) | (1.2) | (30.1) | (2) | (0.1) | (0.1) | (0.1) | (15) | (26) | (14) |
| Clean | 10, 17-18, 21-22 | 4 | 5205 | 0.1 | 2.7 | 67 | 0.1 | 1.1 | 4.9 | 48 | 74 | 26 |
| | *11:00-23:00* | (3) | (1557) | (0.1) | (4.4) | (72) | (0.1) | (1.0) | (5.9) | (36) | (52) | (34) |
| Car emission | *16:00-20:00* of clean days | 3 | 4647 | 0.1 | 1.0 | 69 | 0.2 | 1.2 | 4.2 | 54 | 96 | 43 |
| | | (3) | (1510) | (0.1) | (1.1) | (31) | (0.1) | (0.4) | (2.2) | (13) | (43) | (40) |
| Truck emission | *21:00-23:00* of clean day | 4 | 4671 | 0.1 | 0.6 | 191 | 0.1 | 2.7 | 15.7 | 108 | 132 | 24 |
| | | (3) | (1522) | (0.1) | (0.5) | (89) | (0) | (1.2) | (7.8) | (42) | (20) | (25) |
| Pollution | 12-13, 24-27 | 6 | 4310 | 0.1 | 0.6 | 279 | 0.1 | 2.5 | 9.7 | 130 | 160 | 30 |
| | *8:00-23:00* | (8) | (964) | (0.2) | (0.9) | (111) | (0.1) | (1.1) | (4.6) | (41) | (45) | (9) |
| Firework emission | *8:00-22:00* of 26 | 2 | 4474 | 0.1 | 0.2 | 324 | 0.1 | 2.2 | 10.0 | 135 | 168 | 34 |
| | | (3) | (993) | (0.1) | (0.3) | (40) | (0) | (0.4) | (1.8) | (12) | (12) | (9) |
| Overall | 10-28 | 12 | 4370 | 0.4 | 6.7 | 209 | 0.1 | 1.9 | 7.5 | 102 | 138 | 35 |
| | (8:00-23:00) | (28) | (1229) | (0.9) | (22.6) | (141) | (0.1) | (1.2) | (5.4) | (53) | (60) | (34) |

14. *Lines 215 – 217: Did you also calculate the ice-active surface site density? This might be the better parameter to compare to other studies, as it is a normalized quantity. Moreover, I suggest to compare INP number concentrations only to measurements conducted at the same nucleation temperature.*

[Response]: Thanks! We calculate the ice-active surface site density using the number concentration integrated from particle size distribution below HINC cut-off size in the revised manuscript. Comparison with previous studies conducted at identical or similar conditions are also included. Calculation method of $n_S$ is detailed in Sect. 2.3 between L169-L176.

Deleted comparison with Ardon-Dryer and Levin (2014) and Chen et al. (2021) as suggested.

[Figure]

**Figure R1-4. Ice-active surface site density ($n_S$) for different scenarios. The median (horizontal orange lines), 25$^{th}$ and 75$^{th}$ percentiles (lower and upper boxes), and 10$^{th}$ and 90$^{th}$ percentiles (lower and upper whiskers) are shown.**

L169-L176: "Activation fraction (AF) and ice-active surface site density ($n_S$) were selected as IN activity parameters in this study (Vali et al., 2015; Kanji et al., 2017). AF is the ratio between ice crystal number concentration at HINC outlet (calculated from OPC counts, as stated above) and total particle number concentration at HINC inlet (measured by CPC). $n_S$ is defined as the number of ice-active surface sites per unit surface area of INPs, and allows IN activity inter-comparison between different aerosol species and different studies as a normalized parameter (Hoose and Möhler, 2012; Vali et al., 2015). In this study, total surface area $S$ for poly-disperse ambient particles was firstly derived by assuming particles to be spherical and integrating the particle mobility size distribution (Chen et al., 2021). Dividing $N_{INP}$ by $S$ yields $n_S$ (Connolly et al., 2009; Hoose and Möhler, 2012; Niemand et al., 2012; Vali et al., 2015; Lacher et al., 2018; Bi et al., 2019)."

L236-L241: "$n_S$ during the dust event is also on the same order of magnitude with Bi et al. (2019) and Lacher et al. (2018), as shown in Fig. 5. However, most of the parameterizations obtained from laboratory experiments using Asian dust (AD) samples tend to overestimate $n_S$ by 1-2 orders of magnitude (Connolly et al., 2009; Niemand et al., 2012; Ullrich et al., 2017), except for the size-segregated parameterization proposed by Reicher et al. (2019) that spans from $1.9 \times 10^9$ # m$^{-2}$ for sub-micron to $4.2 \times 10^{10}$ # m$^{-2}$ for super-micron particles. "

15. *Lines 240 – 242: Does the non-relation between particle size and INP concentration during the dust event suggest that the mineralogy of the dust aerosol might be the driving factor for ice nucleation? Do you have any information about the mineralogy of the dust particles?*

[Response]: We agree that mineralogy is a very important driving force for dust IN, as suggested by previous studies (e.g., Atkinson et al., 2013; Kaufmann et al., 2016; Iwata and Matsuki, 2018). Unfortunately, we do not have mineralogy information for our current study. Our next investigations will target collection of filter samples and perform off-line mineralogy analysis, and associate dust mineralogy with online INP concentration measurements. To acknowledge this, we add the following sentence in L266 -L268

L266-L268: "As suggested by previous studies (e.g., Atkinson et al., 2013; Kaufmann et al., 2016; Iwata and Matsuki, 2018), dust mineralogy might be a superior immersion IN driving factor instead of $N_{500}$ and PM$_{10-2.5}$ in the urban environment and worth further exploration."

16. *Line 335: The INP concentration measurements were not performed continuously; e.g., there are several days especially before the 22ⁿᵈ February when no INP concentration measurements were performed.*

[Response]: Deleted "continuously" as suggested.

L371: "In situ observation of $N_{INP}$ and physiochemical properties…"

**Referee #2**

*Summary:*

*Zhang et al. report ice nuclei concentrations ($N_{INP}$) measured under mixed-phase cloud conditions in Beijing over a 19 days that included a variety of aerosol conditions. The authors correlate $N_{INP}$ with several collocated measurements in interpret the sources of INP, including non-refractory composition, black carbon, aerosol mass, and wide-ranging size distributions.*

*The study reports interesting and relevant results that are important to aerosol-cloud studies in the urban environment. The scientific questions under study are within the scope of ACP. The paper is very well organized and written. Literature references are appropriate and demonstrate a strong understanding of the relevant measurements and the sampling environment. The measurement study appears well conceived and executed.*

*The authors clearly demonstrate several negative results that are interesting and important, eg, that INP concentrations do not correlate with urban pollution, traffic emissions, BC, or particles generated by fireworks. However, the role of dust in INP concentrations is more problematic, and the paper needs to address the apparent inconsistencies in the data more clearly.*

*Below I detail 3 concerns that affect some of the paper's conclusions. I encourage the authors to address the major and minor comments, and in particular consider what robust conclusions can be made regarding mineral dust. Once these issues are addressed, the paper will be appropriate for publication in ACP.*

We thank the referee for the recognition of our work. Below, the referee outlines concern on interpretation of correlation analysis and the treatment of measured INP concentration. We have made necessary revisions which we believe address the referee's concerns.

*Major comments:*

1. *The data show that $N_{INP}$ has an inverse relationship to several aerosol quantities like $N_{500}$ and $PM_{10}$-$PM_{2.5}$ that the authors suggest are surrogates for mineral dust. In particular, Fig 5a suggests a robust anti-correlation of $N_{INP}$ and large particles during the dust event. This surprising result is lost in the discussion of correlation coefficients, which are arguably less important here than whether the slope of the correlation plot is positive or negative (or essentially zero). The authors should add text when describing Figs 5, 9, C1, and C2 to discuss whether the quantities are directly or inversely correlated, while using the R^2 values as a guide to the strength of that relationship. Or instead of discussing the slope of the fits, replace the R^2 values with the Pearson's correlation coefficient (R), which is negative for an inverse relationship.*

   *Most importantly, the authors should carefully consider these direct and inverse correlations, and lack thereof, in their conclusions regarding dust aerosol. Specifically, line 352 states, "Our study reveals that mineral dusts, even though present in relatively low number concentration out of the high background particle number concentration, dominate immersion INP population in the urban environment". This statement is not supported by*

*Fig 5, C1a, nor C2, which show an inverse or no relationship between $N_{INP}$ and the aerosol properties chosen as surrogates for dust. Somewhat confusingly, the two campaign-wide correlation plots (C1b and C2b) disagree in the sign of the correlation. These apparent inconsistencies, and particularly the surprising inverse relationships, should be more clearly interpreted in the concluding remarks and abstract (line 27-28). Indeed, for this study it seems that the data are generally inconclusive as to dust's (or large particles') role in ice nuclei, particularly outside of clear dust events.*

[Response]: We thank the referee for the suggestions and agree that the role of mineral dust could be better discussed. $R^2$ has been replaced with Pearson's correlation coefficient ($r$) throughout the manuscript. Corresponding modifications in the text, figures, and captions have been made. Please find more details in the marked-up manuscript.

The different signs of $r$ in Fig. C1b and Fig. C2b was actually attributed to different particle sources and properties in different scenarios. The positive correlation in Fig. C1b is merely caused by the dust event, with all other scenarios collapsed in the lower right corner. $PM_{10}$ and $PM_{2.5}$ show minor difference during the heavy pollution (with $PM_{10-2.5}$ ranges between $30 \pm 9$ µg m$^{-3}$) and other periods while $PM_{10-2.5}$ is notably higher during the East Asian dust event, ranging between $138 \pm 14$ µg m$^{-3}$ (Table 1 and Fig. 3c), which makes the dust event distinctive from other periods. However, the high $N_{INP}$ during the dust event was mitigated by the low $N_{INP}$ during other periods with similar $N_{500}$ level such as clean days, leading to a weak negative correlation ($r = -0.2$) between $N_{INP}$ and $N_{500}$.

Our data consistently suggests that large particles outside the clear dust event, especially during heavy pollution and truck emission periods, are not effective immersion INP under our experiment condition.

To avoid misunderstanding, we have modified the statement on the role of mineral dust during the dust event in the urban environment in the abstract and conclusion as suggested by the reviewer.

L27-L28: "The results indicate a substantial $N_{INP}$ increase during the dust event, although the observation took place at an urban site with high background aerosol concentration."

L388-L389: "Our study reveals that immersion INP population in the urban environment increased substantially during the East Asian dust event.".

2. *It most cases it is statistically incorrect to remove negative values from a set of measurements taken near the limit of detection, LOD (or limit of quantification, LOQ), line 146. The authors correctly state that negative values (and some small positive ones too!) are indistinguishable from zero. However, these 'zeros' represent legitimate results, and they must be included in many instances, for example when calculating a mean value or a correlation with another parameter.*

   *Consider the case where an instrument attempts to measure a property that has a true value of zero. Random statistical noise will result in the measured (signal minus background) being small positive for some samples and small negative for others, centered around zero*

*within the instruments LOD. If you removed all the negative values, your calculated mean will always be artificially positive, where it should actually be zero.*

*Alternately, if a measurement has a clearly defined LOD (eg, a set of filtered-air HINC runs), it is also correct to replace all values <LOD with the value zero. A third option is to replace all values <LOD with the LOD value or LODx0.5. This is a typical solution when logarithm of the data is required. If the authors continue to use correlations in log-space, some variation of this third option is acceptable. Depending on what fraction of the data was removed, including this low/negative/zero measurements may significantly affect the reported correlations.*

[Response]: Thanks for the suggestion! Following the treatment of negative $N_{INP}$ in Lacher et al. (2017), the authors decided to assign a minimum quantifiable concentration (0.26 # L$^{-1}$) for the 15 min HINC run for negative $N_{INP}$. This was done by taking the minimum count possible in HINC OPC >5 μm channel (1 count) and normalizing to the volume of ambient air during a 15 min HINC sampling period (0.26 L min$^{-1}$×15 min). The treatment is detailed between L162-L167.

Including the replaced minimum quantifiable concentration indeed changes the value of derived average $N_{INP}$ and the Pearson's correlation coefficients, but does not affect the conclusion originally drawn in the manuscript. All figures have been redrawn with the updated processed dataset.

L162-L167: "Average ice crystal concentration (equivalent to $N_{INP}$) of the 15-minute measurement period is calculated by firstly subtracting the average background particle counts from measurement counts, and secondly converting particle counts to number concentration using HINC sampling flowrate. The subtraction might produce negative $N_{INP}$ when the signal of OPC during the measurement is undistinguishable from background noise. Therefore, this study reports positive $N_{INP}$ as is, and replaces negative $N_{INP}$ with the minimum quantifiable concentration of OPC (0.26 # L$^{-1}$) following the method in Lacher et al. (2017)."

[Figure]

**Figure R2-1. Correlation analysis between $N_{INP}$ and $N_{500}$ including (orange) and excluding (grey) negative $N_{INP}$.**

3.   *In line 260, the authors consider ammonium secondary material on dust particles acting as a nucleating agent. Although Fig 6 indeed shows a mild correlation between $N_{INP}$ and $m_{ammo}$,*

*"...suggesting that might be associated with $m_{ammo}$ during dust events in the urban environment."* However, the authors do not demonstrate that this correlation is specific to ammonium compared to other secondary aerosol material or to PM0.5 as a whole. The authors should plot or at least report correlations (R or slope and R^2, etc) with sulfate, organic material, and nitrate. If those correlations are noticeably weaker, then this supports the authors' assertion about ammonium. However, if those correlations are similar to $m_{ammo}$, then the conclusion about ammonium salts enhancing nucleation activity is not strongly supported by the data, unless the authors can otherwise demonstrate ammonium's role separate from other chemical components.*

[Response]: Added Table C1 in appendix C. The Pearson's correlation coefficient ($r$) between $N_{INP}$ and $m_{NH4}$ ($r = 0.46$) is substantially higher than with $m_{Org}$ ($r = -0.04$), $m_{NO3}$ ($r = 0.06$), and PM$_{0.5}$ ($r = 0.06$). However, $N_{INP}$ does exhibit a moderate positive correlation with $m_{SO4}$ ($r = 0.36$). Kumar et al. (2018), Kumar et al. (2019) and Whale et al. (2018) examined the impact of several ammonium salts, such as $(NH_4)_2SO_4$, $NH_4NO_3$ and $NH_4Cl$, on the immersion IN activity of microcline and aluminosilicate. They found similar immersion IN activity change in solutions with same ammonium concentration, regardless of the anion species within the solution.

To avoid misunderstanding, we deleted "The synchronized trends of $N_{INP}$ and $m_{ammo}$ suggest that the trace amount of atmospheric ammonium (below 1 µg m$^{-3}$ compared to 5-10 µg m$^{-3}$ on other days, as listed in Table 1) might be internally-mixed with ambient mineral dust particles.", and added "It should be noted that $N_{INP}$ also has a weak positive correlation with $m_{SO4}$ ($r = 0.4$, Table C1) during the dust event, but several studies have confirmed that ammonium content, instead of anion species, is more likely to be the driving force in altering the immersion IN activity of mineral dusts (e.g., Kumar et al., 2018; Whale et al., 2018; Kumar et al., 2019)." between L285 and L288.

L285-L288: "It should be noted that $N_{INP}$ also has a weak positive correlation with $m_{SO4}$ ($r = 0.4$, Table C1) during the dust event, but previous studies have confirmed that ammonium content, instead of anion species, is more likely to be the driving force in altering the immersion IN activity of mineral dusts (e.g., Kumar et al., 2018; Whale et al., 2018; Kumar et al., 2019)."

**Tabel C 1 Pearson's correlation coefficient ($r$) between $N_{INP}$ and selected variables.**

| Measured Values | $N_{500}$ | $N_{1000}$ | $m_{Org}$ | $m_{NH4}$ | $m_{SO4}$ | $m_{NO3}$ | $m_{BC}$ | PM$_{0.5}$ | PM$_{2.5}$ | PM$_{10}$ | PM$_{10-2.5}$ |
|---|---|---|---|---|---|---|---|---|---|---|---|
| Dust Event | -0.36 | -0.25 | -0.04 | 0.46 | 0.36 | 0.06 | -0.09 | 0.06 | 0.05 | -0.23 | -0.49 |
| Overall | -0.29 | 0.02 | -0.32 | -0.32 | -0.34 | -0.37 | -0.30 | -0.37 | -0.19 | 0.22 | 0.70 |

*Minor comments:*

1. *Title. Consider specifying, eg, "...the Beijing urban environment..."*

[Response]: Title changed to "Ice nucleating particles from multiple aerosol sources in the urban environment of Beijing under mixed-phase cloud conditions".

2. *Fig 1/line 105. Describe the TEOM sampling arrangement, or add TEOMs to the figure. Particularly for the TEOMs, were any efforts made to reduce aerosol losses in sampling tubing (gravitational, impaction)? For instance, were driers or transport tubing oriented vertically?*

[Response]: Added TEOM in Fig 1.

[Figure]

**Figure R2-2. Updated Figure 1.**

TEOM was installed at the sampling site, ~3 m away from our sampling inlet. The TEOM inlet was erected vertically to reduce particle loss caused by sharp sampling line edges.

3. *Line 120. Typically, a TSI APS has a total inlet flow of ~1 vlpm. The sheath flow is a closed internal loop. Please correct your text as necessary.*

[Response]: Changed to "The sampling flow rate of APS was 1 LPM."

L132: "The sampling flow rate of APS was 1 LPM."

4. *Section 2.2.2. Although the Aerodyne ACSM is often marketed as a "PM1" instrument, the actual sampling range for their standard inlet range as reported by Ng et al., 2011in the original ACSM instrument paper is dva=75-650nm (the 50% transmission limits). This is equivalent to da<530nm or about "PM0.5", not PM1. The distinction is sometimes irrelevant and is often ignored since submicron aerosol mass is often be restricted to da<530nm. However, for this study Fig2b suggests that much of the true PM1 mass during pollution events is far outside the ACSM size range. Report the actual size range for your ACSM inlet, and replace the "PM1" notation throughout the document with an accurate label.*

*Ng et al., 2011: https://doi.org/10.1080/02786826.2011.560211*

[Response]: This is a great point. Liu et al. (2007) reported that the measured 50% transmission range of ACSM aerodynamic lens is ~ 57-663 nm ($70 < d_{va} < 812$ nm) at 780 hPa (Table 3) and the transmission efficiency dropped mildly outside the range. Fig. 13 in Ng et al. (2011) showed

that ACSM could detect super-micron particle mass. Sun et al. (2013) demonstrated the capability of ACSM to detect non-refractory $PM_1$ in Beijing during wintertime.

To clarify and maintain consistent term usage with previous ACSM papers and Beijing aerosol chemical composition measurement studies (e.g., Sun et al., 2013; Crenn et al., 2015; Xu et al., 2017), we kept the "non-refractory $PM_1$" notation in the manuscript, and added "The 50% transmission efficiency range of ACSM is ~60-660 nm (Liu et al., 2007)." between L136-L137.

L136-L137: "The 50% transmission efficiency range of ACSM is ~60-660 nm (Liu et al., 2007)."

5.  *Line 149. State CPC size range. The ice-active fraction strongly depends on the minimum size of the reference measurement.*

[Response]: Added "$d_{50} = 4$ nm".

L167-L168: "Ambient particle number concentration entering HINC was monitored by a CPC (Model 3775, $d_{50} = 4$ **nm**; TSI Inc.)…"

6.  *Fig 2b. Add more tick labels to left axis.*

[Response]: Tick labels added.

[Figure]

**Figure R2-3. Updated Figure 2.**

7.  *Table 1. N1000 is missing. Define typical start/stop times for noted dates.*

[Response]: Typical measurement time added. Extra rows and columns have been added to provide more useful information to readers.

**Updated Table 2 The date, number concentrations of immersion INP ($N_{INP}$) and ambient particle ($N_{CPC}$), activation fraction (AF), ice-active surface site density ($n_S$), number concentrations of particles ranging from 500 nm to 1.5 μm ($N_{500}$) and 1000 nm to 1.5 μm ($N_{1000}$), as well as mass concentrations of BC particles ($m_{BC}$) and ammonium salt ($m_{ammo}$), PM$_{2.5}$ and PM$_{10}$, and PM$_{10-2.5}$ for each scenario. The numbers are average values, and numbers in parentheses denote one standard deviation ($\sigma$) from the average.**

| Scenario | Date of Feb. and time periods | $N_{INP}$ (# L$^{-1}$) | $N_{CPC}$ (# cm$^{-3}$) | AF×10$^3$ (%) | $n_S$×10$^{-8}$ (# m$^{-2}$) | $N_{500}$ (# cm$^{-3}$) | $N_{1000}$ (# cm$^{-3}$) | $m_{BC}$ (μg m$^{-3}$) | $m_{ammo}$ (μg m$^{-3}$) | PM$_{2.5}$ (μg m$^{-3}$) | PM$_{10}$ (μg m$^{-3}$) | PM$_{10-2.5}$ (μg m$^{-3}$) |
|---|---|---|---|---|---|---|---|---|---|---|---|---|
| Dust | 21 | 112 | 3364 | 3.6 | 90.0 | 26 | 0.1 | 0.4 | 0.6 | 57 | 194 | 137 |
|  | *12:00-18:00* | (34) | (952) | (1.2) | (30.1) | (2) | (0.1) | (0.1) | (0.1) | (15) | (26) | (14) |
| Clean | 10, 17-18, 21-22 | 4 | 5205 | 0.1 | 2.7 | 67 | 0.1 | 1.1 | 4.9 | 48 | 74 | 26 |
|  | *11:00-23:00* | (3) | (1557) | (0.1) | (4.4) | (72) | (0.1) | (1.0) | (5.9) | (36) | (52) | (34) |
| Car emission | *16:00-20:00 of clean days* | 3 | 4647 | 0.1 | 1.0 | 69 | 0.2 | 1.2 | 4.2 | 54 | 96 | 43 |
|  |  | (3) | (1510) | (0.1) | (1.1) | (31) | (0.1) | (0.4) | (2.2) | (13) | (43) | (40) |
| Truck emission | *21:00-23:00 of clean day* | 4 | 4671 | 0.1 | 0.6 | 191 | 0.1 | 2.7 | 15.7 | 108 | 132 | 24 |
|  |  | (3) | (1522) | (0.1) | (0.5) | (89) | (0) | (1.2) | (7.8) | (42) | (20) | (25) |
| Pollution | 12-13, 24-27 | 6 | 4310 | 0.1 | 0.6 | 279 | 0.1 | 2.5 | 9.7 | 130 | 160 | 30 |
|  | *8:00-23:00* | (8) | (964) | (0.2) | (0.9) | (111) | (0.1) | (1.1) | (4.6) | (41) | (45) | (9) |
| Firework emission | *8:00-22:00 of 26* | 2 | 4474 | 0.1 | 0.2 | 324 | 0.1 | 2.2 | 10.0 | 135 | 168 | 34 |
|  |  | (3) | (993) | (0.1) | (0.3) | (40) | (0) | (0.4) | (1.8) | (12) | (12) | (9) |
| Overall | 10-28 | 12 | 4370 | 0.4 | 6.7 | 209 | 0.1 | 1.9 | 7.5 | 102 | 138 | 35 |
|  | (8:00-23:00) | (28) | (1229) | (0.9) | (22.6) | (141) | (0.1) | (1.2) | (5.4) | (53) | (60) | (34) |

8. *Line 190. Clarify that you are inferring dust composition and therefore the dust event. Specifically, PM10>>P2.5 indicates that large particles are present. What is actually implied/inferred is that PM10>>PM2.5 is due to mineral dust aerosol. Reference an appropriate Asian dust PM10/PM2.5 if that is helpful.*

[Response]: Changed to "…implying that large particles with $d_a$ ranging between 2.5 μm and 10 μm were present, which were most likely to be mineral dust particles (e.g., Park et al., 2004; Hoffmann et al., 2008; Rai et al., 2021)."

L210-L212: "…except for Feb. 21$^{st}$ afternoon, when there was a significant increase of PM$_{10}$ mass concentration as highlighted by the orange shading in Fig. 3, implying that large particles with $d_a$ ranging between 2.5 μm and 10 μm were present, which were most likely to be mineral dust particles (e.g., Park et al., 2004; Hoffmann et al., 2008; Rai et al., 2021)."

9. *Line 217. Delete "an"*

[Response]: Deleted.

10. *Line 235. Reword or delete the sentence "It would be worthwhile...". The suggested course of action is confusing because the authors actually go on to explore this.*

[Response]: Rephased.

L262-L266: "To explore the connection between PM$_{10-2.5}$ and $N_{INP}$ in the urban environment, correlation analysis between $N_{INP}$ and PM$_{10-2.5}$ data collected during the dust event is also conducted using OLS linear regression. The correlation between $N_{INP}$ and PM$_{10-2.5}$ at -30 °C ($r =$ -0.5, Fig. C1) during the dust event in this study suggests that PM$_{10-2.5}$ has a moderate negative

correlation with $N_{\mathrm{INP}}$ in the urban environment, which is stronger yet not statistically significant."

11. *Line 278-281. Why might the DeMott 2015 dust INP parameterization vastly overestimate the measured INP here? Might "large" particles be something other than dust? Would dust likely be coated with secondary material like sulfate and organics (in addition to ammonium salts)? Will these coatings deactivate dust to the nucleation mechanism under study (add any appropriate refs)?*

[Response]: Thanks for the suggestion! Mixing with sulfate and ammonium salts could impair the immersion IN activity of mineral dusts (Cziczo et al., 2009; Eastwood et al., 2009; Chernoff and Bertram, 2010; Niedermeier et al., 2011; Tobo et al., 2012; Augustin-Bauditz et al., 2014) depending on the amount of ammonium ion present. Added discussion on the potential explanation of the overestimation, and suggest that future parameterizations should take the effects of atmospheric processes into consideration.

L306-L316: "Such overestimation on $N_{\mathrm{INP}}$ during the dust event by previous parameterizations might be caused by omission of mineral dust chemical composition and mixing state change during transportation (e.g., Tobo et al., 2010; Wang et al., 2014; Li et al., 2016; Tang et al., 2016; Wu et al., 2020). Previous laboratory experiments suggest that even though mixing with organics might not affect the immersion IN activity of mineral dusts (Tobo et al., 2012; Wex et al., 2014; Kanji et al., 2019), mixing with sulfuric acids (Cziczo et al., 2009; Eastwood et al., 2009; Chernoff and Bertram, 2010; Niedermeier et al., 2011; Tobo et al., 2012; Augustin-Bauditz et al., 2014) and ammonium and sulfate salts (Cziczo et al., 2009; Iwata and Matsuki, 2018; Kumar et al., 2018; Whale et al., 2018; Kumar et al., 2019) could suppress the immersion IN activity of mineral dusts to different degrees. However, current parameterizations could represent the upper limit of atmospheric INP number concentration in global models, and we suggest that future parameterizations should include the influence of atmospheric processes (such as photo-oxidation and gaseous species condensation) on mineral dust IN activity to achieve more realistic prediction."

12. *Line 304. As written, it is unclear if this is a valid comparison since total BC concentrations might be very different in Schill. State the BC concentrations or the active fractions for both studies.*

[Response]: Added AF and $n_{\mathrm{S}}$ comparison.

L339-L342: "AF and $n_{\mathrm{S}}$ reported by Schill et al. (2016) were about one order of magnitude smaller than measured AF and $n_{\mathrm{S}}$ associated with vehicle emission periods (Table 1 and Fig. 5)."

13. *Line 321. Unclear wording "with R2 between...". Again, it is important that 5b shows anti-correlation.*

[Response]: Statement revised.

L358-L360: "$\log_{10}(N_{INP})$ exhibits a weak negative correlation with $\log_{10}(N_{500})$ in Fig. 6b ($r = -0.2$). As shown in Fig. 10b, the OLS linear regression results further suggest that $N_{INP}$ is likely to be independent of $m_{BC}$ during heavily polluted days ($r = 0$)."

14. *Line 344. Awkward phrase "synchronized variation". Suggest replacing with ""a weak positive correlation" or similar.*

[Response]: Changed.

L380-L381: "During the dust event, $N_{INP}$ exhibit a moderate positive correlation with $m_{ammo}$ ($r = 0.5$) and a moderate negative correlation with $PM_{10-2.5}$ ($r = -0.5$)"

15. *Line 344. "...and $N_{INP}$ exhibited slight dependence on PM10-2.5." Clarify that it's an inverse dependence!*

[Response]: Elaborated. Please see our response to last comment.

16. *Fig A1. Clarify the "BG" measurement period. (Is it the same as "clean" in Table 1?)*

[Response]: Specified the measurement period in the caption.

"BG" refers to the night before the dust event.

L394-L395: "Figure A1. Aerodynamic diameter ($d_a$) size distribution of particles before (00:00-06:00, Feb. 21$^{st}$; BG) and during the dust event (11:00-18:00, Feb. 21$^{st}$; Dust)."

17. *Fig A3. Show altitudes of the trajectories as colors or as a separate graph. (Is the air over the desert near the surface or aloft?)*

[Response]: Altitude added. The altitude suggests that the air was aloft.

[Figure]

Figure R2-4. Updated Figure A3.